# Harnessing insulin- and leptin-induced oxidation of PTP1B for therapeutic development

Navasona Krishnan[1], Christopher A. Bonham[1], Ioana A. Rus[1,2], Om Kumar Shrestha[1], Carla M. Gauss[1], Aftabul Haque[1], Ante Tocilj[1,3,4], Leemor Joshua-Tor [1,3,4] & Nicholas K. Tonks[1]

The protein tyrosine phosphatase PTP1B is a major regulator of glucose homeostasis and energy metabolism, and a validated target for therapeutic intervention in diabetes and obesity. Nevertheless, it is a challenging target for inhibitor development. Previously, we generated a recombinant antibody (scFv45) that recognizes selectively the oxidized, inactive conformation of PTP1B. Here, we provide a molecular basis for its interaction with reversibly oxidized PTP1B. Furthermore, we have identified a small molecule inhibitor that mimics the effects of scFv45. Our data provide proof-of-concept that stabilization of PTP1B in an inactive, oxidized conformation by small molecules can promote insulin and leptin signaling. This work illustrates a novel paradigm for inhibiting the signaling function of PTP1B that may be exploited for therapeutic intervention in diabetes and obesity.

[1] Cold Spring Harbor Laboratory, 1 Bungtown Road, Cold Spring Harbor, NY 11724, USA. [2] Graduate Program in Genetics and Medical Scientist Training Program, Stony Brook University, 100 Nicolls Road, Stony Brook, NY 11794, USA. [3] W. M. Keck Structural Biology Laboratory, Cold Spring Harbor Laboratory, 1 Bungtown Road, Cold Spring Harbor, NY 11724, USA. [4] Howard Hughes Medical Institute, Cold Spring Harbor Laboratory, 1 Bungtown Road, Cold Spring Harbor, NY 11724, USA. Navasona Krishnan, Christopher A. Bonham and Ioana A. Rus contributed equally to this work. Correspondence and requests for materials should be addressed to N.K.T. (email: tonks@cshl.edu)

Obesity and type 2 diabetes, together with their cardio-vascular complications, pose a serious threat to public health in the 21st century. According to the American Diabetes Association, almost 10% of the US population has diabetes, with 86 million Americans over the age of 20 in a prediabetic state. Furthermore, the WHO highlights that 422 million people worldwide were afflicted with diabetes in 2014, compared to 108 million in 1980. These numbers are expected to grow in the next few decades, with a prevalence of 600 million projected by 2035[1,2]. Type 2 diabetes, which is caused by insulin resistance resulting in loss of normal glucose homeostasis, accounts for >90% of all diabetes. This disease used to be referred to as "adult- or maturity-onset" diabetes, but is now also becoming more prevalent throughout the population, including in children.

**Fig. 1** Molecular characterization of PTP1B-scFv45 interaction. **a** Overlay of PTP1B (gray) and TCPTP (pale blue) crystal structures, highlighting residues that are changed in generating mutant forms (-mut1 and -mut2) of the enzymes. **b** Increasing concentrations of recombinant HA-tagged scFv45 were incubated with oxidized wild-type (-wt) or mutant (-mut) forms of PTP1B (top panel) and oxidized wild-type or mutant forms of TCPTP (bottom panel). The protein complexes were precipitated with anti-HA-agarose beads and equivalent amounts of input, supernatant, and eluate were subjected to immunoblot analysis. Data are representative of at least three independent experiments ($n = 3$). **c** Michaelis–Menten plots for wild-type (circle), -mut1 (square), or -mut2 (triangle) forms of PTP1B (left) and TCPTP (right) are shown. Radiolabeled $^{32}$PY-RCML was used as substrate to measure phosphatase activity. Individual data points represent the mean obtained from three separate experiments ($n = 3$). **d** Wild-type and mutant forms of PTP1B were incubated with increasing concentrations of $H_2O_2$ and phosphatase activity was measured using DiFMUP in the absence (black bar) and presence (gray bar) of the reducing agent TCEP (5 mM). **e** Oxidized PTP1B-wt (circle), PTP1B-mut1 (square), or PTP1B-mut2 (triangle) were incubated with increasing concentration of scFv45 (0–1000 nM) and phosphatase activity was measured using DiFMUP as substrate in the presence of the reducing agent TCEP (5 mM). Data represent mean ± s.e.m. from three separate experiments ($n = 3$; **d**, **e**)

This illustrates that therapeutic options for treating diabetes and obesity are inadequate, and effective approaches to counter the disease are urgently needed.

The ability to modulate signal transduction pathways selectively holds enormous therapeutic potential. The protein tyrosine phosphatase PTP1B, which is a negative regulator of insulin and leptin signaling, is a highly validated target for therapeutic intervention in diabetes and obesity[3,4]. The *PTPN1* gene, which encodes PTP1B, is located in 20q13, a genomic region that is linked to insulin resistance and diabetes in human populations from different geographical origins. More than 20 single nucleotide polymorphisms (SNPs) that are associated with increased risk of type 2 diabetes have been identified within the *PTPN1* gene[5]. Whole-body deletion of PTP1B in mice resulted in increased insulin sensitivity and improved glucose tolerance[6,7]. These animals were found to be lean, hypersensitive to leptin and resistant to diet-induced obesity[8,9]. In addition to improved glucose homeostasis, hepatic PTP1B deletion was found to contribute to positive lipid metabolic changes in both liver and circulation[10]. The serum cholesterol levels in these animals were found to be lower than in control littermates, even when subjected to a high fat diet for a prolonged period of time. Conversely, insulin resistance can result in hepatic fat accumulation, which is a significant contributor to non-alchoholic steatohepatitis (NASH)[10,11]. In addition, liver biopsies from patients with NASH also revealed an increase in PTP1B levels[12]. In light of such data, considerable interest has grown in the potential of PTP1B as a therapeutic target for treating diabetes and obesity. Consequently, there have been substantial programs in industry focused on developing small molecule inhibitors of this phosphatase. All of these programs followed standard procedures of looking for active site-directed inhibitors; however, these efforts have been frustrated by technical challenges arising from the chemical properties of the PTP active site. The end result has been that although it was possible to generate potent, specific, and reversible inhibitors of PTP1B, such molecules were highly charged and thus of low oral bioavailability and limited drug development potential[13,14]. Hence, alternative strategies are needed to develop drugs against this highly validated therapeutic target.

In one such approach, we wanted to harness a physiological regulatory mechanism in which tyrosine phosphorylation-dependent signaling, such as in response to insulin, triggers the localized production of reactive oxygen species, in particular $H_2O_2$, leading to inactivation of PTP1B through reversible oxidation of the phosphatase. This represents a mechanism for fine-tuning the signaling response[15,16]. Previously, we used phage display technology to generate a library of conformation-sensor antibodies that recognize epitopes unique to the reversibly oxidized form of PTP1B (PTP1B-OX), which are not displayed in the active, reduced form of the enzyme[17]. We characterized these conformation-sensor antibodies, such as scFv45, which stabilized the reversibly oxidized, inactive form of PTP1B. Following expression in cells, scFv45 trapped the oxidized form of PTP1B, preventing its reduction and reactivation, which resulted in enhanced and sustained insulin-induced signal transduction. These conformation-sensor antibodies displayed remarkable specificity. For example, scFv45 did not bind to the reduced form of PTP1B, nor did it recognize TCPTP (either in the reduced or oxidized form), which displays ~75% sequence identity to PTP1B in its catalytic domain. Although this profound specificity would be an asset to a drug candidate, such antibodies are unlikely to be developed therapeutically. Nevertheless, a small molecule that mimics the effects of these antibodies could have therapeutic potential.

In this study, we have identified the binding site for scFv45 on PTP1B-OX and present a molecular basis for the specificity of this interaction. We observed that, in addition to insulin signaling, scFv45 enhanced the activation of signaling in response to leptin. Furthermore, we present the identification and characterization of candidate small molecule inhibitors that mimic the effects of the scFv45 antibody on signal transduction. The characterization of one such small molecule, chelerythrine, illustrates the exciting possibility that stabilization of the oxidized, inactive form of PTP1B with appropriate therapeutic small molecules may be exploited to improve insulin and leptin signaling. Overall, this study provides proof-of-principle for a novel approach to PTP-based drug development that circumvents the challenges posed by the chemistry of the PTP active site.

## Results

**Molecular basis for the interaction between scFv45 and PTP1B-OX.** Reversible oxidation of PTP1B is accompanied by profound conformational changes in the protein, particularly in the active site[18,19]. Although PTP1B and TCPTP are closely related in sequence, there are residues that are unique to each enzyme, including on the surface of the proteins close to the active site (Fig. 1a). We used mutagenesis and subsequent binding assays with scFv45 to identify residues in PTP1B that are important for the interaction between the two proteins. Several mutant forms of PTP1B were generated following comparison of its sequence and structure with that of TCPTP, and substitution of those residues from TCPTP into the corresponding PTP1B sequence. Conversely, chimeric TCPTP mutants containing residues substituted from PTP1B were also generated. We focused on two mutant constructs, a control designated PTP1B-mut1, and PTP1B-mut2 (Fig. 1a), and identified three important residues in the binding interface between PTP1B and scFv45. Using a co-immunoprecipitation assay, we observed that, compared to the wild-type enzyme, PTP1B-wt, the binding interaction between scFv45 and reversibly oxidized PTP1B-mut2 (L37F/K39E/K41R) was abrogated (Fig. 1b). Conversely, introduction of these three residues (Leu37, Lys39, and Lys41) into TCPTP was sufficient to promote binding between scFv45 and reversibly oxidized TCPTP-mut2 (F39L/E41K/R43K) (Fig. 1b). Similar substitution of S28H/F30Y/C32H to generate PTP1B-mut1 did not interfere with binding of scFv45, nor did the converse mutations in TCPTP-mut1 gain binding (Fig. 1b). Using the artificial substrate $^{32}$PTyr-RCML to assay phosphatase activity, we observed that the specific activity of PTP1B-mut1 and PTP1B-mut2 was comparable to that of PTP1B-wt (Fig. 1c). Furthermore, PTP1B-mut1 and PTP1B-mut2 were reversibly oxidized by $H_2O_2$ and reactivated by the reducing agent TCEP in a similar manner to wild-type PTP1B (Fig. 1d). Upon incubation of the reversibly oxidized forms of PTP1B-wt and PTP1B-mut1 with scFv45, the reduction and reactivation of the enzyme was impaired in a dose-dependent manner (Fig. 1e); in contrast, scFv45 did not attenuate the reduction and reactivation of reversibly oxidized PTP1B-mut2. Taken together, these data demonstrate that Leu37, Lys39, and Lys41 in the loop comprising residues 36–41 in PTP1B-OX are critical for scFv45 binding independently of an effect on phosphatase activity or reversible oxidation.

Following mutation of the catalytic Cys215 and adjacent Ser216 to alanine (PTP1B-CASA), PTP1B adopts a conformation that is identical to that of the reversibly oxidized sulfenyl-amide form (PTP1B-OX)[17]. Therefore, this mutant allows us to investigate scFv45 binding on a homogenous background, ensuring that any sample heterogeneity caused by incomplete oxidation during sulfenyl-amide formation is eliminated. In a complementary approach to the analysis by mutagenesis described above, we examined the interaction of scFv45 with PTP1B-CASA using analytical size-exclusion chromatography.

We observed that unlike PTP1B-CASA, PTP1B-CASA-mut2 was unable to associate with scFv45, whether in excess or at limiting molar ratios (Fig. 2a; Supplementary Fig. 1a, b). Moreover, when similar experiments were performed with TCPTP-CASA and TCPTP-CASA-mut2, we observed a gain-of-association in TCPTP-CASA-mut2, compared to TCPTP-CASA, suggesting that TCPTP-CASA-mut2 adopts a similar architecture to that of PTP1B-OX and PTP1B-CASA (Supplementary Fig. 2). These data are in agreement with the binding studies using oxidized and mutant forms of PTP1B and TCPTP (Fig. 1), and support the utility of CASA mutants, and further mutants thereof, for detailed analysis of scFv45 association. Similarities in the reversible oxidation and catalytic properties of PTP1B and PTP1B-mut2 (Fig. 1b–e), together with the uniform chromatographic profiles of PTP1B-CASA and PTP1B-CASA-mut2 (Fig. 2a), suggest that the inability for PTP1B-CASA-mut2 to associate with scFv45 was

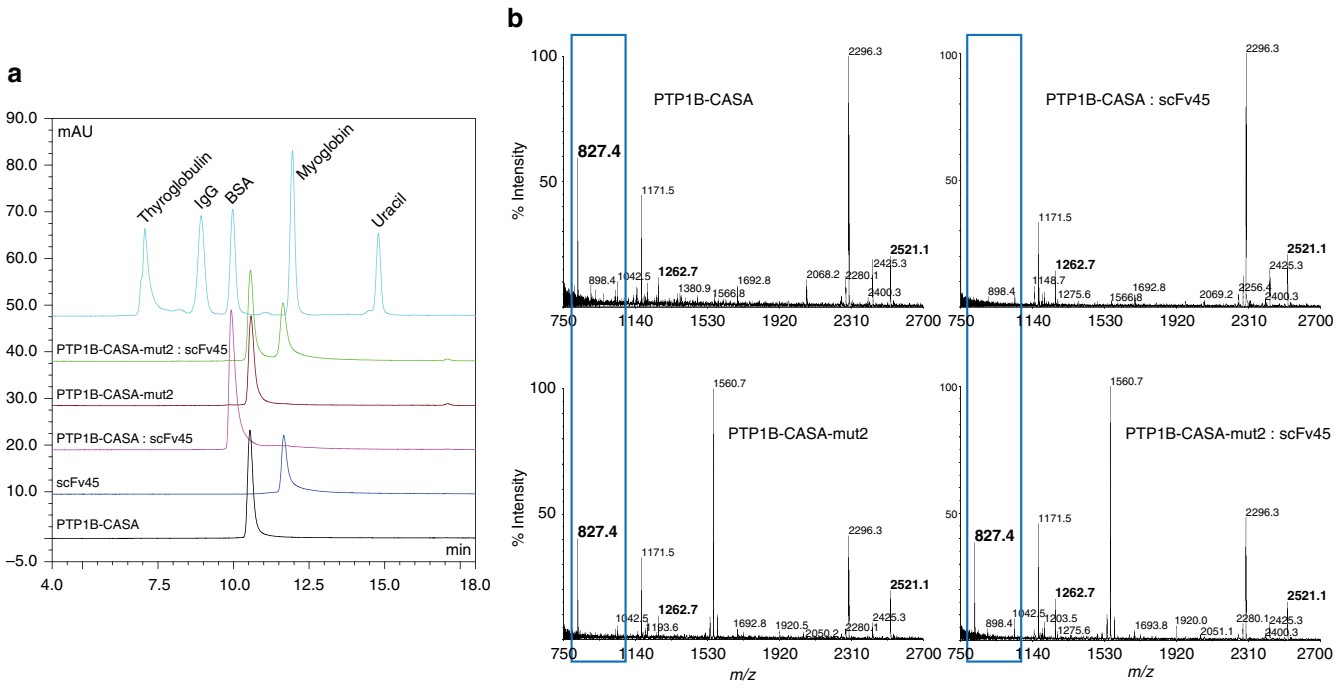

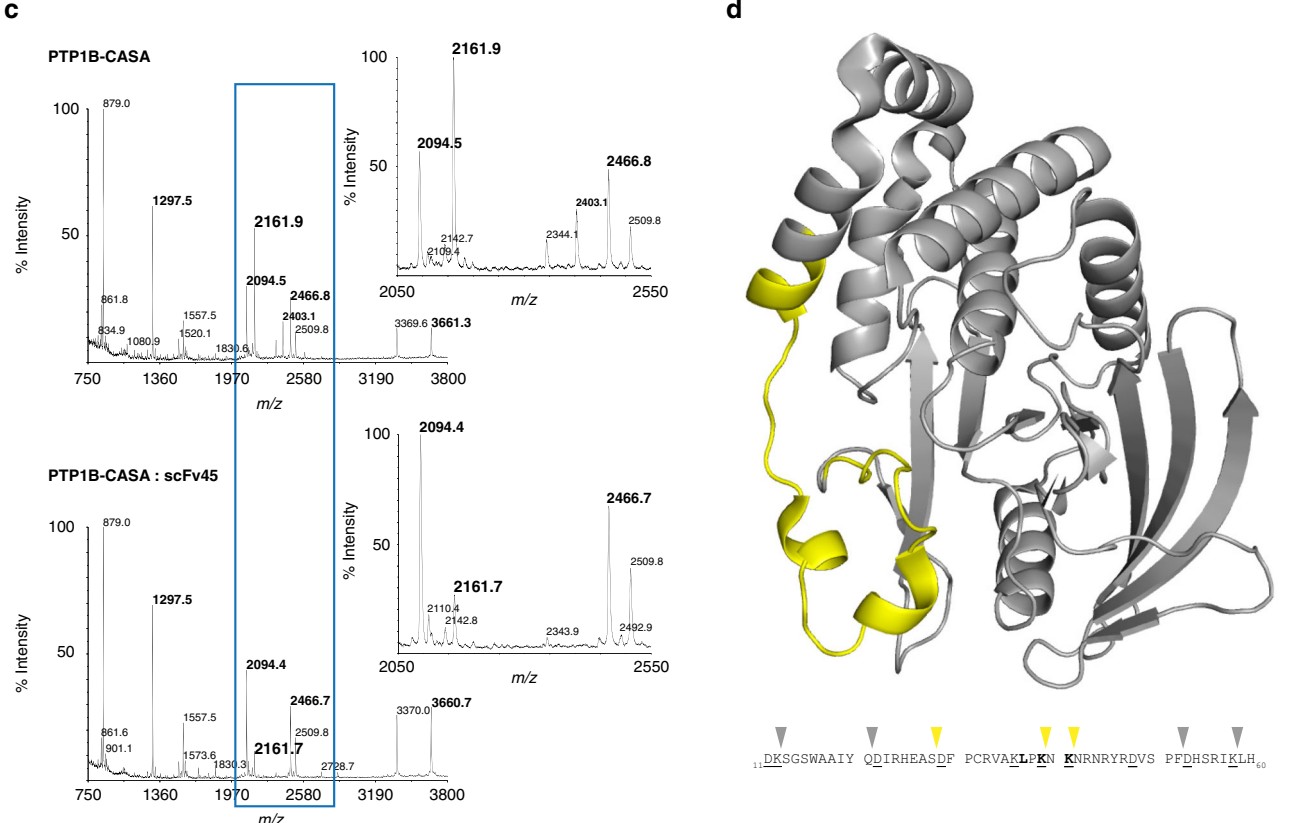

likely due to a lack of key binding determinants rather than a consequence of gross structural perturbations.

We used a combination of unbiased (by proteolytically targeting Asp residues) and biased (by proteolytically targeting Lys residues) mass spectrometry-based structural proteomics to characterize this interaction further and define residue boundaries for segments of PTP1B involved in association with scFv45. Upon complex formation between PTP1B-CASA and scFv45, we observed that proteolytic cleavage at Asp28 by endoproteinase AspN was nearly undetectable, whereas in the absence of scFv45 or in mixtures of PTP1B-CASA-mut2 with scFv45, cleavage at this site was readily observed ($m/z$ 827.4; Fig. 2b; Supplementary Fig. 3a, b). Accessibility of proximal, amino-terminal cleavage sites at Asp22 and Asp53 ($m/z$ 2521.1 and 1262.7), as well as those in the carboxyl-terminus ($m/z$ 1171.5 and 2296.3), were independent of PTP1B-CASA association with scFv45 (Fig. 2b; Supplementary Fig. 3a, b). Also, we observed a marked reduction in the relative accessibility of amino-terminal proteolytic cleavage sites towards endoproteinase LysC in complexes of PTP1B-CASA and scFv45, specifically at Lys39 and Lys41 ($m/z$ 2403 and 2161, respectively; Fig. 2c; Supplementary Fig. 4a, b). Cleavage at amino-terminal sites Lys12 and Lys58 ($m/z$ 1557 and 1773, respectively), as well as in the carboxyl-terminus ($m/z$ 879 and 3370), was independent of association with scFv45. Overall, these data suggest regional boundaries of the binding interface from Ile23 to Phe52 (Fig. 2c, d; Supplementary Figs. 3 and 4). Complementary analysis by protein electrophoresis supports that the observed variations among these initial proteolytic events sampled were the result of scFv45 association with PTP1B-CASA and not the consequence of exhaustive digestion inducing protein unfolding or degradation of either PTP1B or scFv45 (Supplementary Fig. 5).

To provide further insights into the molecular details of the interaction between scFv45 and the reversibly oxidized form of PTP1B, we determined the crystal structure of scFv45 at 1.6 Å resolution (Table 1). This single-chain variable fragment (scFv) antibody contains only the variable light ($V_L$) and variable heavy ($V_H$) chains of the antigen-binding fragment ($F_{ab}$) connected by an 18 amino acid linker[20]. The structure consists of the $V_H$ and $V_L$ chains, with no electron density observed for the linker region, in accordance with other reported scFv X-ray crystal structures[21]. Within scFv45, acidic residues dominate the hypervariable complementarity-determining region (Fig. 3a). This feature is shared among all the scFvs that were identified as being inhibitory to reduction and reactivation of PTP1B-OX[17]. It is interesting to note also that PTP1B-OX presents a high density of basic residues near the active site, which have an important role in scFv45 binding, with Arg47 fully exposed in comparison to reduced PTP1B (Figs. 1 and 2).

Attempts to produce crystals of a purified complex of scFv45 and PTP1B-OX were unsuccessful. Therefore, we performed an in silico docking approach using Cluspro software with Antibody Mode option, with svFv45 as a receptor and PTP1B-OX as a ligand to generate a low-energy structure of the complex. From the model it was clear that acidic residues dominated the complementarity-determining region (CDR) of scFv45; Asp89 and Asp93 of scFv45 formed hydrogen-bonding interactions with basic residues Arg47 and Lys41 of PTP1B-OX, respectively (Fig. 3b). By mapping the regional boundaries of the binding interface that were determined by our structural proteomic approach onto this model, we noted strong corroboration between the individual datasets (Figs. 2d and 3b). We also generated electrostatic maps for PTP1B-OX, scFv45 and scFv20 using the Adaptive Poisson–Boltzmann Solver (APBS) tool in PyMol (Fig. 3c). The acidic residues of scFv45 in the CDR region, which are critical for interaction with PTP1B-OX, were found to be lacking in scFv20, suggesting a molecular explanation for the lack of interaction observed between PTP1B-OX and scFv20[17].

**Small molecule inhibitors that stabilize PTP1B-OX.** We have demonstrated that we can generate reversibly oxidized pools of PTP1B in vitro, with reproducible reactivation by reducing agents such as TCEP (Fig. 1d). Therefore, our approach has been to adapt this assay, which we used to screen for conformation-sensor scFvs[17], now to screen for small molecules that act through a similar mechanism. Our primary screening assay tested the ability of small molecule inhibitors to antagonize the reduction and reactivation of the oxidized conformation of PTP1B, similar to that of scFv45 (Fig. 4a). Therefore, in the presence of small molecules that do not stabilize the oxidized form of PTP1B, we expect to observe restoration of enzymatic activity, and dephosphorylation of the fluorogenic substrate DiFMUP to the level observed in controls treated with reducing agent alone; in contrast, with any small molecules that stabilize the oxidized conformation of PTP1B, the restoration of enzymatic activity would be impaired.

PTP1B was titrated with varying concentrations of $H_2O_2$ to optimize the concentration at which the enzyme would be stably, reversibly oxidized for a period of 30–60 min. This is critical as the overall time to complete the assay in an HTS (High-Throughput Screen) set up could vary between 30 and 60 min and within that period of time it is essential to maintain standard conditions in all assays. PTP1B was then reversibly oxidized in large batches with $H_2O_2$. The oxidized enzyme was incubated without (high signal) and with (low signal) scFv45 for 5 min, following which the enzyme was reactivated with TCEP. The high and low signals were used to calculate the Z-prime value of 0.75 for the assay. The fluorescence signal was robust and the difference between the high and low signal was ~25-fold (Supplementary Fig. 6).

The assay was validated using the LOPAC library (Library of Pharmacologically Active Compounds), which contains 1280 compounds (Fig. 4a). Two compounds in the library were found to stabilize partially the oxidized conformation; addition of the reducing agent TCEP did not increase the fluorescence intensity

**Fig. 2** Oxidation-induced conformation of PTP1B exposes an amino-terminal binding interface targeted by scFv45. **a** Analytical size-exclusion chromatography of PTP1B-CASA and PTP1B-CASA-mut2 in the absence or presence of 1.1-fold molar excess of scFv45. **b** Mass spectra of PTP1B-CASA and PTP1B-CASA-mut2 in the absence or presence of 1.1-fold molar excess of scFv45, digested with AspN endoprotease (1:500 w/w) for 30 min. Peptide masses defining regional boundaries of PTP1B-CASA proteolytic protection are shown in bold, with those directly protected being highlighted (blue rectangle). **c** Mass spectra of PTP1B-CASA in the absence or presence of 1.1-fold molar excess scFv45, digested with LysC endoprotease (1:750 w/w) for 30 min, mixed with internal peptide standards (~42.5 fmol). Masses of internal peptide standards are shown in bold, with those peptide masses of PTP1B-CASA directly protected from proteolysis being enlarged and in bold. Scaled insets are of the highlighted (blue rectangle) spectral regions. All data are representative of at least two independent experiments ($n \geq 2$; **a–c**). **d** Three-dimensional crystal structure of PTP1B-CASA, highlighting the regional boundaries corresponding to the segment protected from proteolysis upon association with scFv45 (Ile23—Phe52). Below is the amino acid sequence of the N-terminal portion of PTP1B in which residues that were (yellow triangles) or were not (grey triangles) protected from proteolysis are highlighted. Theoretical AspN and LysC cleavage sites (underlined) and mutated residues in PTP1B-CASA-mut2 (bold) are shown

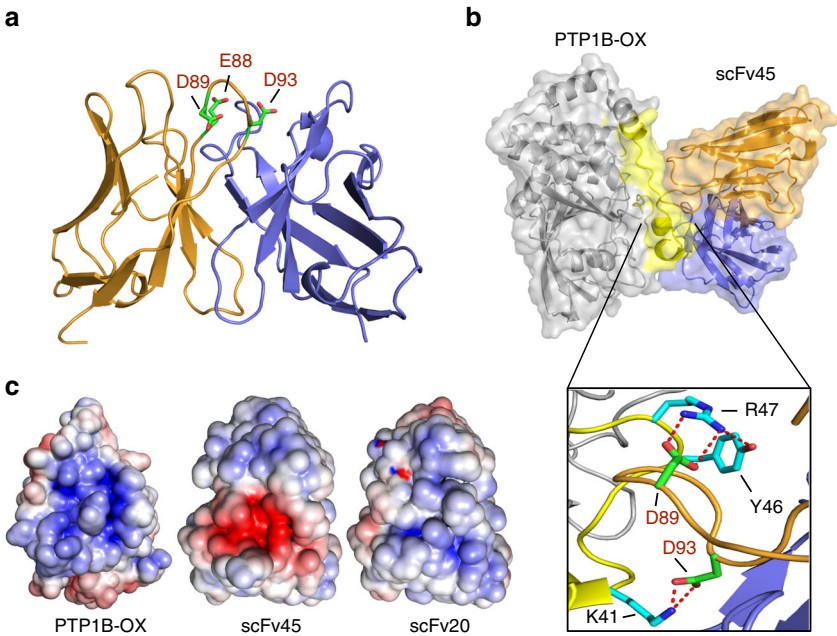

**Fig. 3** Crystal structure of scFv45 and molecular explanation for scFv45 specificity for PTP1B-OX. **a** Crystal structure of scFv45 highlighting acidic residues Asp89, Glu88, and Asp93 in the complementarity-determining region (CDR). For representative numbering of scFv45 residues, the first amino acid denotes the residue directly following the OmpA cleavage site. **b** Molecular modeling of the PTP1B-OX:scFv45 complex, with PTP1B-OX shown in gray, and the regional boundaries of the scFv45-binding interface determined by our structural proteomic approach shown in yellow. The inset shows H-bonding between residues of PTP1B-OX (K41, Y46, and R47) and the acidic residues in the CDR of scFv45. **c** Surface electrostatic potential of PTP1B-OX (left), scFv45 (middle), and scFv20 (right)

to the same extent as that of the positive control (without any inhibitor or scFv45). With both hits, sanguinarine and levamisole, TCEP was only able to restore enzymatic activity partially (Fig. 4a). Similar results were obtained with a distinct reducing agent, dithiothreitol (DTT). By titrating varying concentrations of sanguinarine and levamisole against PTP1B-OX we were able calculate an $IC_{50}$ of 25 μM for sanguinarine (Fig. 4b) and 36 μM for levamisole (Fig. 4c). These compounds had no effect on the activity of the reduced form of PTP1B. To highlight further the significance of the hits, and to confirm that their inhibitory effects were not due to non-specific denaturation, we demonstrated that their effects were reversible by initiating enzymatic assays following increasing incubation times of compound-bound PTP1B-OX in the presence of reducing agent (Fig. 4d, e).

Of the two hits, sanguinarine was found to be the more potent at stabilizing PTP1B-OX, therefore, we focused our attention on this molecule. In searching for analogs with improved inhibitory potency, we identified chelerythrine, chelidonine, palmatine, and protopine, which contained a similar benzophenanthridine scaffold, but with minor changes to the functional groups (Figs. 4f, g; Supplementary Fig. 7). Of these compounds, chelerythrine was shown to be the most potent at stabilizing PTP1B-OX, with an $IC_{50}$ of 5 μM (Fig. 4f) and at least 100-fold preference over reduced PTP1B. The effects of chelerythrine were reversible (Supplementary Fig. 8) and direct binding of chelerythrine to PTP1B-CASA was demonstrated by ITC (Supplementary Fig. 9). The structurally similar molecule, protopine was ineffective in inhibiting either the reduced form of PTP1B, or the reduction and reactivation of the oxidized enzyme (Fig. 4g). These data illustrate that PTP1B-OX can be selectively targeted by small molecules and suggest the beginnings of a structure–activity relationship.

**Mechanism of PTP1B-OX stabilization by chelerythrine.** Three crystal structures were selected for a molecular modeling

approach to identify and compare potential predicted binding sites on PTP1B. One (PDB ID: 1OEM) represented PTP1B-OX, whereas the other two represented the reduced form of PTP1B with the active site in either the open (PDB ID: 1T4J) or closed (PDB ID: 1A5Y) conformation[17,22,23]. The general conformation of PTP1B-OX resembles that of the open form of PTP1B. A key difference in the structures is that Tyr46 does not form a hydrogen bond with Ser216 in PTP1B-OX, but instead is oriented away from the active site[17,18]. Each structure was treated with Schrödinger's SiteMap module, which identifies potential binding sites based on the donor, acceptor and hydrophobic character of regions within the structure, and the extent to which a ligand may form substantive interactions with these regions. Five top-scoring sites were found for each conformation, one of which was unique to PTP1B-OX. This unique site encompassed a loop composed of residues 36–46, which contains Tyr46 and is the segment of the protein that we identified as important for the selective recognition of PTP1B-OX by the antibody scFv45 (Figs. 1–3).

The two compounds that were confirmed hits from the LOPAC 1280 library screen, sanguinarine and levamisole, were docked using Schrödinger's Glide module with the unique potential binding site of PTP1B-OX that was predicted by SiteMap. Interestingly, sanguinarine, but not levamisole, appeared to form a π-stacking interaction with Tyr46, perhaps accounting for its stronger ability to prevent reduction of PTP1B-OX to the active form of the enzyme (Supplementary Fig. 10a, b). The sanguinarine analog chelerythrine displayed improved potency, with a fivefold lower $IC_{50}$, although the only structural difference from sanguinarine is the rotatable methoxy groups (Fig. 4f). When chelerythrine was docked into the unique site identified by SiteMap, as described above, it also formed a π-stacking interaction with Tyr46 (Fig. 5a). Residues Lys36, Lys41, and Asn44, which we had identified previously as potentially contributing to the binding of sanguinarine, also appeared to contact chelerythrine (Fig. 5a; Supplementary Fig. 10a). A

restrained minimization was conducted on the docked model of chelerythrine with PTP1B-OX using Schrödinger's Macromodel. The results of the final treatment indicated an additional hydrogen bond between Arg45 and the methoxy groups of

chelerythrine (Fig. 5a). We performed a mutational analysis to test this model and to understand further the interaction between the compound and the protein. We found that when both residues Lys36 and Lys41 were mutated to alanine, there was a

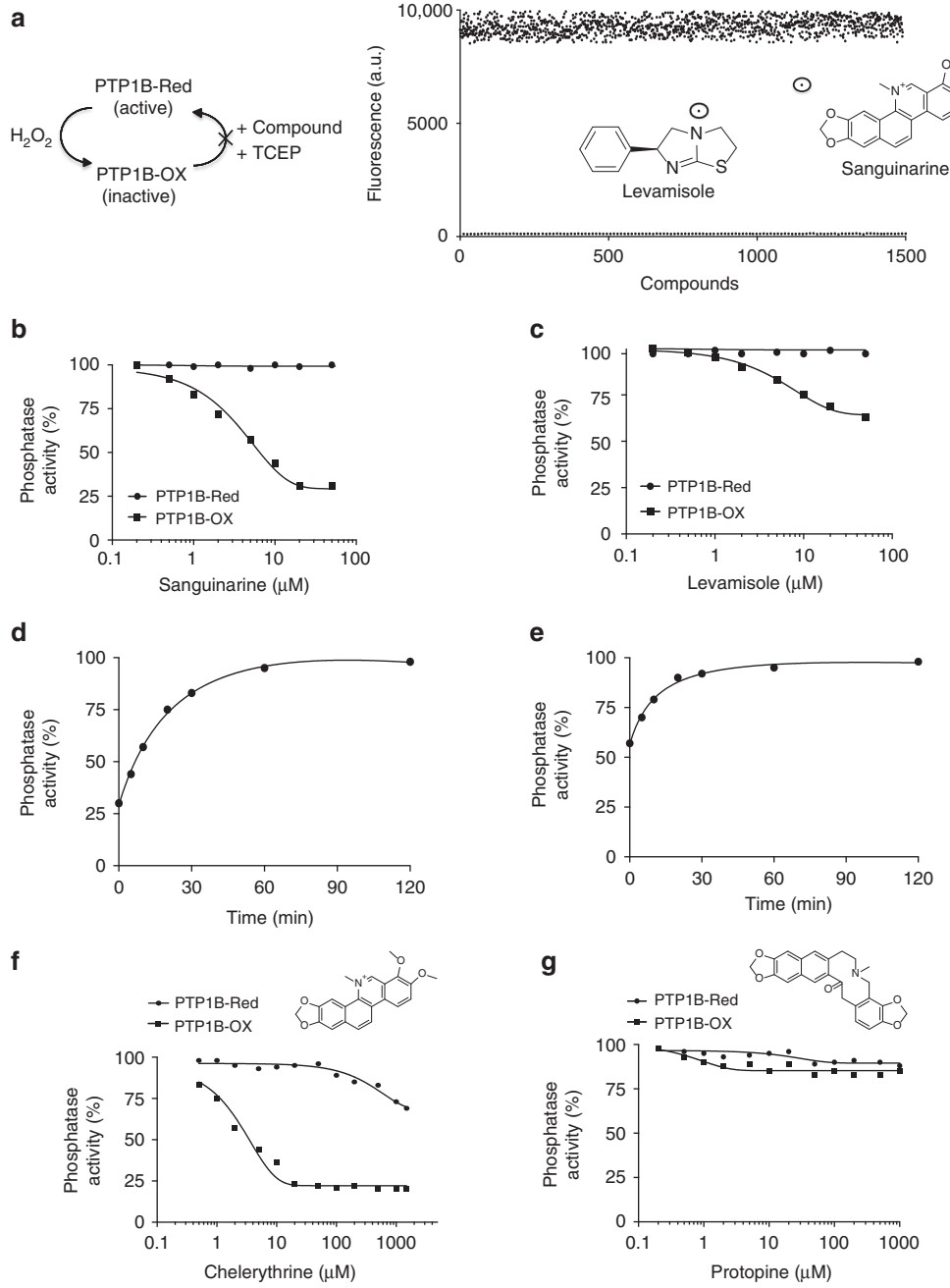

**Fig. 4** Identification and characterization of small molecules that mimic the effect of scFv45. **a** Reversibly oxidized PTP1B (PTP1B-OX) was generated by incubation with $H_2O_2$ in vitro, desalted, then assayed (20 nM) in the presence and absence of individual members of the LOPAC library. In the right panel, each dot represents an activity measurement. With two compounds, the reduction and reactivation of PTP1B-OX by TCEP was attenuated. The structures of these two compounds, levamisole and sanguinarine, are provided. Data are representative of at least two independent experiments ($n = 2$). **b** Increasing concentration of sanguinarine was tested against reduced (circle) and oxidized (square) forms of PTP1B, using DiFMUP as substrate. **c** Increasing concentration of levamisole was tested against reduced (circle) and oxidized (square) PTP1B, using DiFMUP as substrate. **d** PTP1B-OX (200 nM) was incubated with sanguinarine (25 μM) for 5 min and the complex was diluted 100-fold and the activity was monitored for 120 min continuously. **e** PTP1B-OX (200 nM) was incubated with levamisole (40 μM) for 5 min and the complex was diluted 100-fold and the activity was monitored for 120 min continuously. **f** Increasing concentration of chelerythrine, an analog of sanguinarine, was tested against reduced (circle) and oxidized (square) forms of PTP1B, using DiFMUP as substrate. A $k_i$ of 5 μM was calculated for PTP1B-OX and no apparent inhibition of the reduced form was detected. **g** Increasing concentration of protopine, an analog of sanguinarine, was tested against reduced (circle) and oxidized (square) forms of PTP1B, using DiFMUP as substrate. The compound did not inhibit the reduced or the oxidized form of the protein. Individual data points represent the mean obtained from three separate experiments ($n = 3$; **b**–**g**)

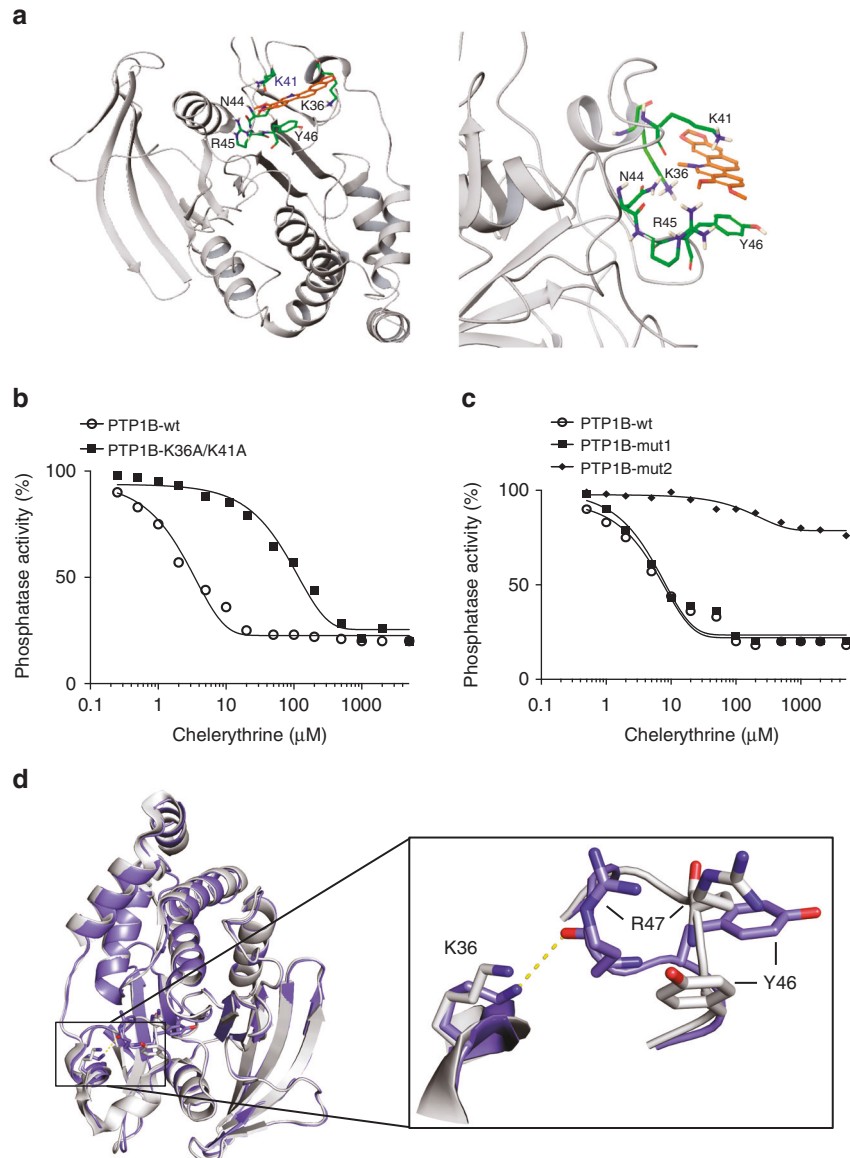

**Fig. 5** Chelerythrine and scFv45 inhibit reactivation of PTP1B-OX by a similar mechanism. **a** Molecular modeling of chelerythrine binding to PTP1B-OX, with critical residues for the interaction highlighted on the right. **b** Oxidized PTP1B-wt (circle), PTP1B-K36A/K41A (square) were incubated with increasing concentrations of chelerythrine (0–10 mM) and phosphatase activity was measured using DiFMUP as substrate in the presence of the reducing agent TCEP (5 mM). **c** Oxidized PTP1B-wt (circle), PTP1B-mut1 (square), or PTP1B-mut2 (diamond) were incubated with increasing concentrations of chelerythrine (0–10 mM) and phosphatase activity was measured using DiFMUP as substrate in the presence of the reducing agent TCEP (5 mM). Individual data points represent the mean obtained from three separate experiments ($n = 3$; **b**, **c**). **d** Overlay of crystal structures of the reduced (blue) and oxidized (gray) form of PTP1B. The inset shows the change in the position of residues K36, Y46, and R47

marked change in the $IC_{50}$ (Fig. 5b). Wild-type PTP1B-OX displayed an $IC_{50}$ of 5 µM for chelerythrine, whereas inhibition of the K36A/K41A PTP1B-OX mutant was attenuated, with an $IC_{50}$ of 75 µM for the same compound (Fig. 5b). We also tested single mutants K36A, K41A, and N44A, but individually the differences in the inhibitory potency were less noticeable.

These data suggest a putative binding site for chelerythrine that overlaps with the site that was found to be critical for scFv45 binding (Figs. 1–3). This is also supported by the observation that scFv antibodies with affinities closer to that of chelerythrine were able to compete with the small molecule for binding to PTP1B-OX (Supplementary Fig. 11). The oxidized form of PTP1B-mut2, in which Leu37, Lys39, and Lys41 in PTP1B were mutated to the equivalent residues in TCPTP (L37F/K39E/K41R), did not bind to scFv45, scFv67, and scFv106 (Fig. 1; Supplementary Fig. 11).

Interestingly, chelerythrine was also unable to inhibit reduction and reactivation of the oxidized form of PTP1B-mut2 (Fig. 5c). In addition, chelerythrine did not prevent complex formation between PTP1B-OX and a thioredoxin substrate trapping mutant[24], suggesting that its effects were not simply due to restricting access of reducing agents to the catalytic cysteine in PTP1B (Supplementary Figs. 12 and 13). By comparing the structures of the oxidized and reduced conformations of PTP1B, we found that the residue Lys36 in PTP1B-OX moves and is unable to H-bond to Arg47, unlike in reduced PTP1B. Thus, the orientation of Lys36 may also be critical for chelerythrine binding to PTP1B-OX (Fig. 5d). These data suggest that the chelerythrine-binding site in PTP1B overlaps with that of scFv45 and provide an explanation for the observed selectivity of the compound for the oxidized form of the phosphatase.

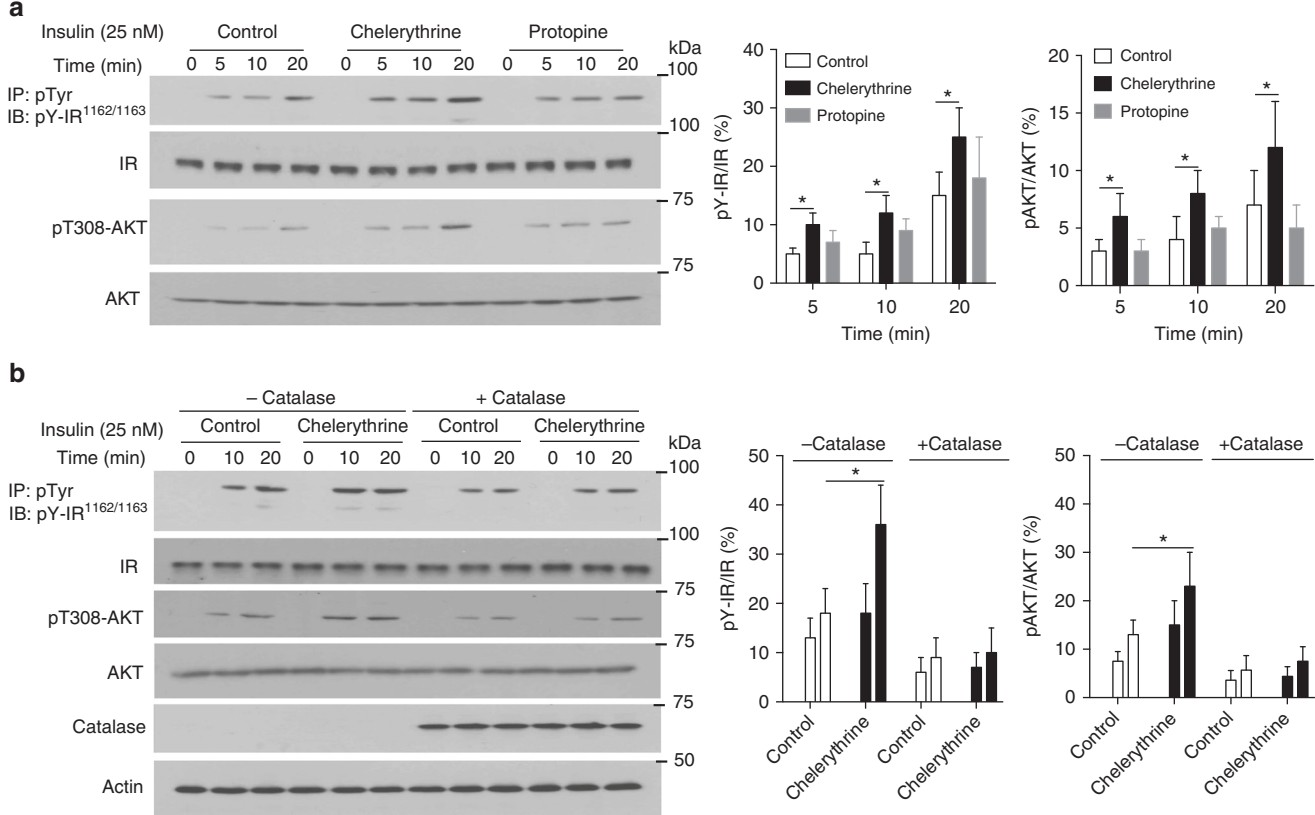

**Fig. 6** Chelerythrine enhanced insulin signaling in a redox-dependent manner. **a** Untreated cells and cells treated with chelerythrine (2 μM) or protopine (2 μM) for 1 h, were stimulated with insulin (25 nM) for the indicated times, following which cells were lysed and lysates were used to study insulin signaling by immunoblot analysis. **b** Cells that transiently express catalase or vector control were untreated or treated with chelerythrine (2 μM) and stimulated with insulin (25 nM) for the indicated times, following which cells were lysed and lysates were used to study insulin signaling by immunoblot analysis. All immunoblots are representative of three independent experiments ($n = 3$). Immunoblots were quantified using ImageJ software. Quantitation includes the data from all experiments and is presented as the mean ± s.e.m. Statistical analysis was performed using two-sided Students $t$-test (*$p < 0.05$)

**Chelerythrine enhanced insulin signaling in cells**. We demonstrated that chelerythrine sequestered reversibly oxidized PTP1B and prevented reactivation of the enzyme. Furthermore, we showed that a structurally related compound, protopine, had no apparent effect on reduction and reactivation of PTP1B-OX. Therefore, we tested the ability of the two compounds to potentiate insulin signaling in 293T cells. When cells treated with chelerythrine were stimulated with insulin, enhanced and sustained phosphorylation of the insulin receptor β-subunit (IR-β) was observed (Fig. 6a; Supplementary Fig. 14). We also tested the effect of the compounds on downstream signaling; using a phospho-specific antibody that recognizes the phosphorylation of Thr308 in AKT, we observed that chelerythrine enhanced insulin-induced activation of AKT. In contrast, when cells were treated with protopine, there was no significant effect on phosphorylation of the insulin receptor or activation of AKT in response to insulin stimulation (Fig. 6a). Upon overexpression of the $H_2O_2$-degrading enzyme catalase, the stimulatory effect of chelerythrine on IR-β phosphorylation and AKT activation was ablated (Fig. 6b). This important control demonstrated that insulin-induced ROS production, and presumably PTP1B oxidation, was essential for the effects of chelerythrine on insulin signaling. Furthermore, overexpression of either PTP1B-wt or PTP1B-mut2 in 293T cells attenuated insulin-induced phosphorylation of the β-subunit of the insulin receptor to a similar extent (Supplementary Fig. 15). In the presence of chelerythrine, we observed elevated insulin signaling in control and PTP1B-wt expressing cells; in contrast, cells expressing PTP1B-mut2 were insensitive to chelerythrine

(Supplementary Fig. 15). The data are consistent with PTP1B-OX as the target of chelerythrine for its effects on insulin signaling in these cells.

**Leptin-induced oxidation of PTP1B augmented signaling**. Upon binding to its receptor, leptin induces phosphorylation and activation of the protein tyrosine kinase JAK2, which initiates downstream signaling responses[25], whereas PTP1B has been identified as an antagonist of JAK2 function[26]. In healthy liver, hepatic stellate cells are in a quiescent state; however, following injury, these cells become activated into myofibroblasts that produce extracellular matrix components, such as collagen, resulting in fibrosis[27]. There have been reports that leptin signaling may contribute to this fibrotic activity and that ROS production is an important component of this signaling function[28]. Therefore, we used hepatic stellate cells as a model system to test for leptin-induced oxidation of PTP1B.

We confirmed that following treatment of hepatic stellate cells with leptin there was elevated phosphorylation of JAK2 (Supplementary Fig. 16a). Furthermore, leptin administration led to the production of ROS, in a dose- and time-dependent manner (Supplementary Fig. 16b). Using scFv45 to immunoprecipitate PTP1B-OX, we demonstrated that leptin induced the oxidation of PTP1B in hepatic stellate cells in a dose-dependent manner (Fig. 7a). Ectopic expression of scFv45 led to enhanced phosphorylation of JAK2 in response to leptin (Fig. 7b). Furthermore, this stimulatory effect was abrogated by treatment

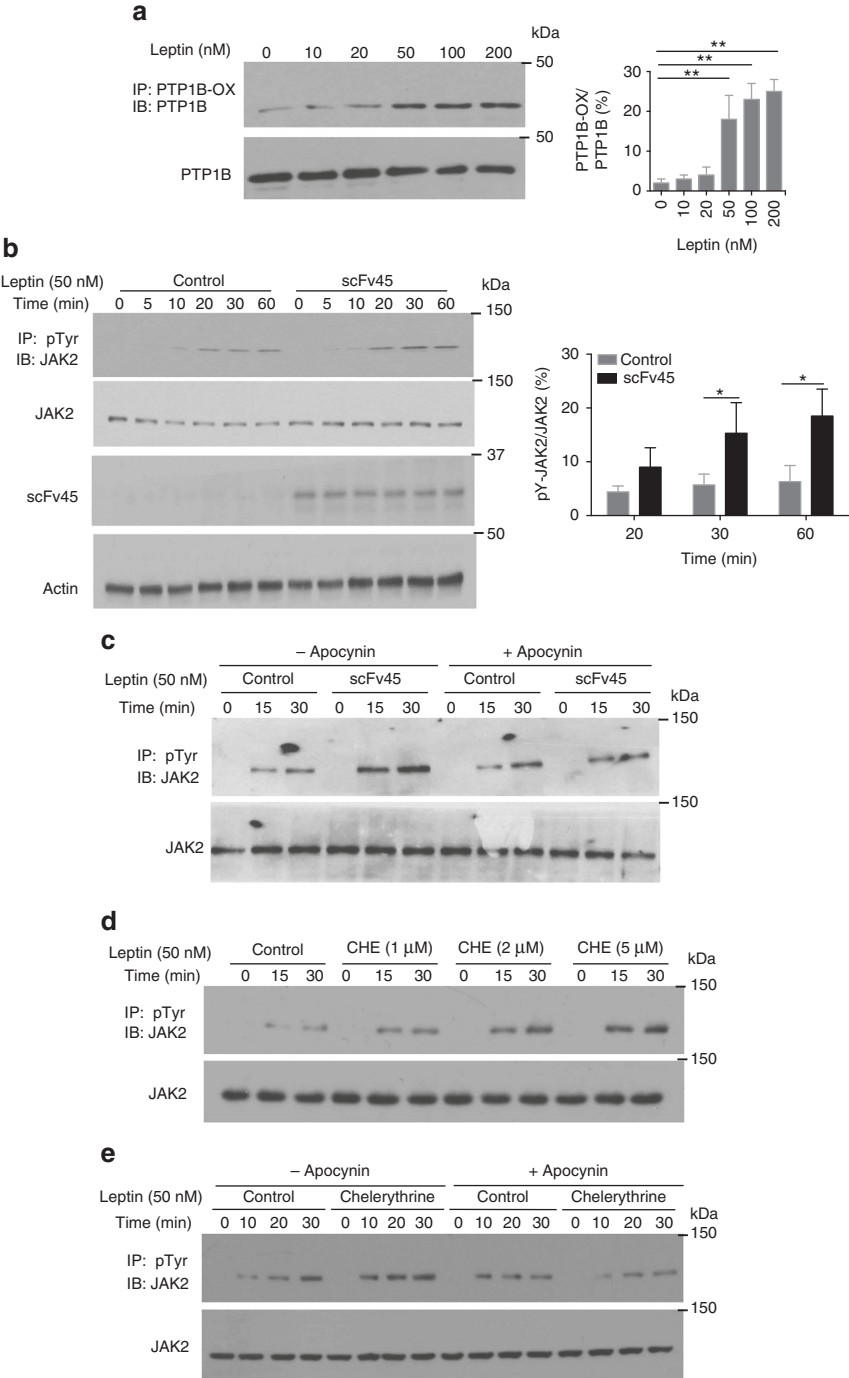

**Fig. 7** Chelerythrine enhanced leptin signaling in a redox-dependent manner. **a** Hepatic stellate cells were stimulated with varying concentration of leptin (0–200 nM), cells were lysed and lysates were used to immunoprecipitate oxidized PTP1B using HA-tagged scFv45. **b** Hepatic stellate cells that transiently express scFv45 or vector control were stimulated with leptin for varying lengths of time (0–60 min) following which cells were lysed and lysates were used to study leptin signaling by immunoblot analysis. **c** Hepatic stellate cells that transiently express scFv45 or vector control were incubated without or with apocynin for 60 min, following which cells were stimulated with leptin for varying lengths of time (0–30 min), lysed, and lysates were used to study leptin signaling by immunoblot analysis. **d** Hepatic stellate cells were treated with varying concentrations of chelerythrine (0–5 μM), stimulated with leptin for varying lengths of time (0–30 min), then lysed, and lysates were used to study leptin signaling by immunoblot analysis. **e** Hepatic stellate cells were incubated with or without apocynin and chelerythrine for 60 min, following which cells were stimulated with leptin for varying lengths of time (0–30 min), then were lysed, and lysates were used to study leptin signaling by immunoblot analysis. All immunoblots are representative of three independent experiments ($n = 3$). Immunoblots were quantified using ImageJ software. Quantitation includes the data from all experiments and is presented as the mean ± s.e.m. Statistical analysis was performed using two-sided Students $t$-test (*$p < 0.05$, **$p < 0.01$)

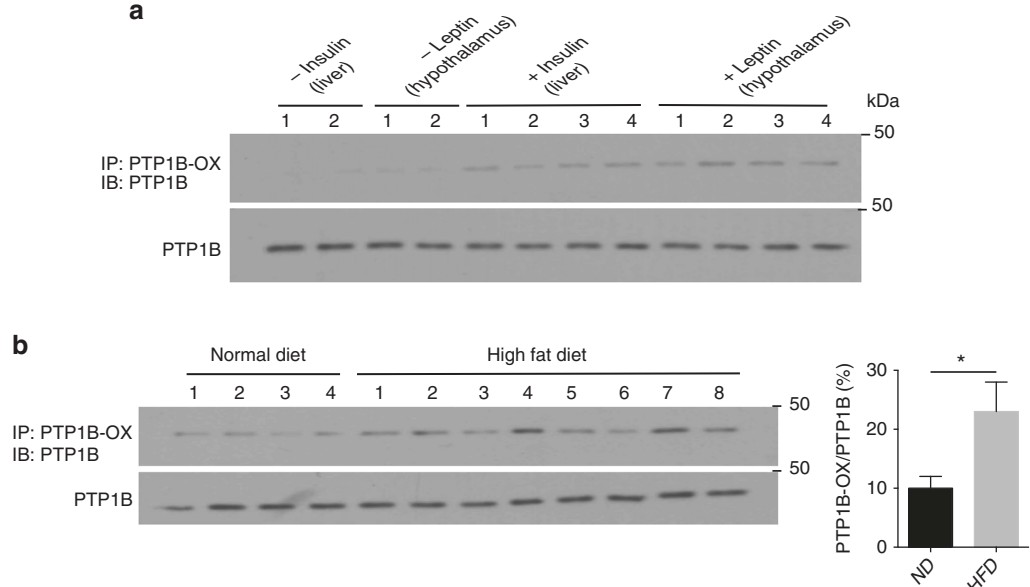

**Fig. 8** Stabilization of oxidized PTP1B in diet-induced obese mice. **a** Oxidation of PTP1B was measured in liver and hypothalamus samples obtained from mice following the administration of insulin and leptin, respectively ($n = 4$ in each group) using a pulldown assay with the PTP1B-OX-specific intrabody scFv45. **b** Oxidation of PTP1B was measured in liver samples obtained from mice fed normal or high fat diet (HFD) using a pulldown assay with the PTP1B-OX-specific intrabody scFv45 ($n = 4$ or more in each group). All immunoblots are representative of tissue lysates generated from at least three independent animals for each condition ($n = 3$). Immunoblots were quantified using ImageJ software. Quantitation includes all independent data from the respective experiments and is presented as the mean ± s.e.m. Statistical analysis was performed using two-way ANOVA (*$p < 0.05$)

with apocynin, which as an inhibitor of NOX (NADPH Oxidase) attenuates ROS production (Fig. 7c). With this in mind, we tested the effects of cheleryrthrine in stellate cells to assess the extent to which it mimicked the effects of scFv45. We observed that chelerythrine treatment induced a dose-dependent increase in tyrosine phosphorylation of JAK2 (Fig. 7d) and that these effects were attenuated by inclusion of apocynin (Fig. 7e). These results illustrate that, similar to insulin signaling, the response to leptin may be fine-tuned by reversible oxidation and inactivation of the negative regulator PTP1B. Furthermore, they suggest that by stabilizing PTP1B-OX, chelerythrine may be able both to improve glucose homeostasis and reverse obesity induced by a high fat diet.

**Chelerythrine improved metabolism in high fat diet-fed mice.** To test this hypothesis, we used high fat diet-fed C57Bl6/J mice to examine the effects of chelerythrine on metabolism in animals. As a first step, we established that PTP1B was present in an oxidized state. Using scFv45 as a probe, we demonstrated that administration of insulin or leptin to chow-fed animals led to elevated oxidation of PTP1B in liver and hypothalamus, respectively (Fig. 8a). Furthermore, the level of PTP1B-OX was elevated ~2.5-fold in high fat diet-fed animals compared to those on a normal chow diet (Fig. 8b).

We treated high fat diet- and chow diet-fed mice with chelerythrine, protopine or saline. In contrast to saline, chelerythrine-treated, high fat diet-fed mice started losing weight within a week of treatment. The weight loss continued for about 14 days, after which no significant decrease in body weight was observed. Overall, we observed ~3% decrease in body weight with chelerythrine (Fig. 9a). Furthermore, no significant change in body weight was observed with chelerythrine-treated mice fed a chow diet, suggesting the weight loss observed is a consequence of improved metabolism in the obese model. Interestingly, weight loss was not observed in protopine-treated mice. This is consistent with the inability of protopine to elicit any inhibitory

effect on either reduced or oxidized PTP1B, and suggests that the effects of chelerythrine resulted from stabilization of PTP1B-OX. Furthermore, we demonstrated that, in contrast to saline or protopine treatment, chelerythrine improved glucose tolerance and insulin sensitivity in glucose and insulin tolerance tests (Fig. 9b, c; Supplementary Fig. 17).

These data suggest that chelerythrine led to enhanced insulin signaling in the high fat diet-fed mice. To examine this further, we measured the effect of the compound on tyrosine phosphorylation of the insulin receptor β-subunit and activation of downstream signaling, through phosphorylation of AKT in the liver. We observed that there was a marked increase in IR-β phosphorylation following chelerythrine treatment, which was not observed with saline or protopine (Fig. 9d). Consistent with this, we observed enhanced AKT phosphorylation with chelerythrine treatment, indicating improved insulin signaling in response to the compound (Fig. 9d). Considering also the important role of leptin in the control of glucose homeostasis and obesity[29,30], we tested the effects of the compounds on leptin signaling in the hypothalamus. Similar to the improvement in insulin signaling, we observed that chelerythrine treatment, but not saline or protopine, led to enhanced phosphorylation of JAK2 (Fig. 9e).

## Discussion
Dietary and lifestyle changes that began to appear in the latter part of the last century have contributed to a global epidemic in obesity, which underlies a dramatic increase in cases of type 2 diabetes and associated cardiovascular complications. The resulting loss of normal control of blood glucose levels in these conditions represents a huge challenge for healthcare systems worldwide. A greater awareness of the "prediabetic" state has prompted many to embrace changes in diet and exercise that help to restore normal control of glucose homeostasis before the development of full-blown diabetes; however, on its own this is insufficient and therapeutic options are required. Metformin is a first line medication that addresses elevated blood glucose,

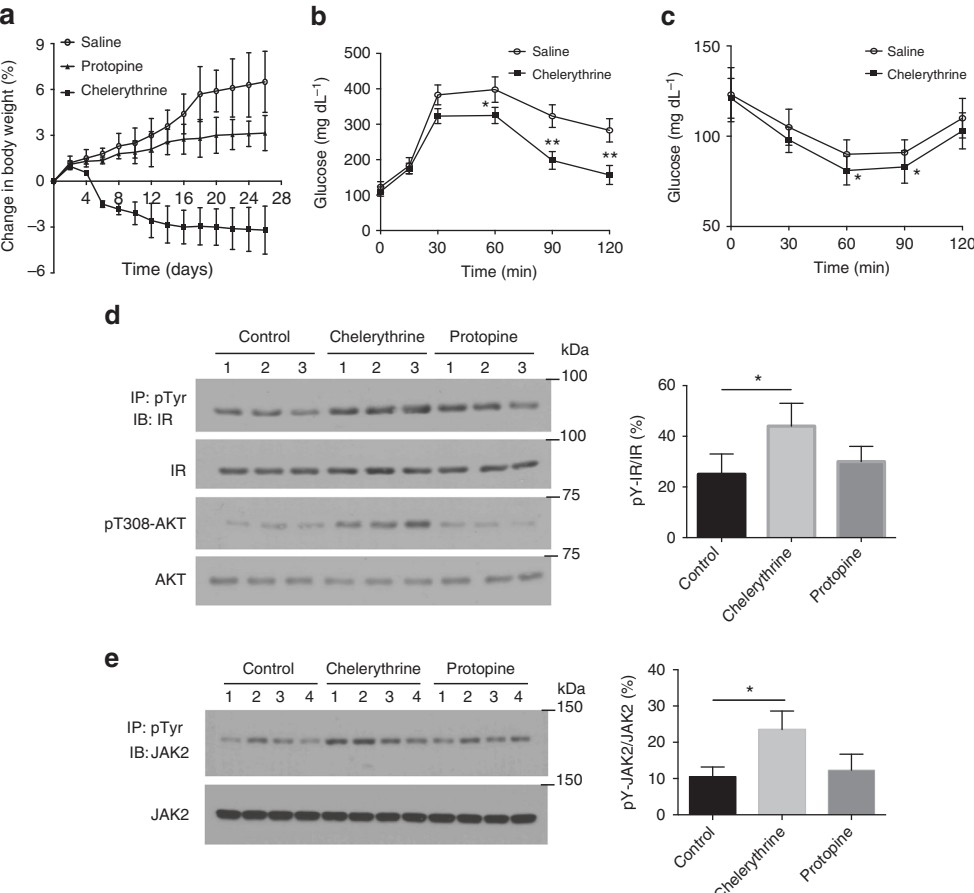

**Fig. 9** Chelerythrine improved metabolism and promoted insulin and leptin signaling in mice. **a** Body weight of male mice (C57bl6/J) fed high fat diet and treated with chelerythrine (3 mg kg$^{-1}$), protopine (3 mg kg$^{-1}$) or saline was monitored from the beginning of the treatment (10 weeks of age) until the study was terminated (14 weeks of age). Data represent the mean ± s.e.m. ($n = 10$ in each group). **b** HFD fed 14-week-old male mice treated with saline or chelerythrine were administered D-glucose (2 mg g$^{-1}$ body weight), and blood glucose was monitored over the defined time. Data represent the mean ± s.e.m. ($n = 10$ in each group) and statistical analysis was performed using two-way ANOVA (*$p < 0.05$, **$p < 0.01$). **c** HFD fed 14-week-old male mice treated with saline or chelerythrine were administered insulin (0.75 mU g$^{-1}$ body weight), and blood glucose was monitored over the defined time. Data represent mean ± s.e.m. ($n = 10$ in each group) and statistical analysis was performed using two-way ANOVA (*$p < 0.05$). **d** Representative immunoblots showing insulin-induced tyrosine phosphorylation of IR-β and phosphorylation of AKT, and their corresponding total protein levels, in liver tissue lysates from 14-week-old male mice treated with saline, chelerythrine or protopine. For insulin stimulation, animals were treated with insulin (0.75 mU g$^{-1}$, i.p.) for 15 min. **e** Representative immunoblots showing leptin-induced tyrosine phosphorylation of JAK2 and corresponding total JAK2 levels in the hypothalamus tissue lysates from 14-week-old male mice treated with saline, chelerythrine or protopine. For leptin stimulation, animals were treated with leptin (1 mg kg$^{-1}$, s.c.) for 15 min. All immunoblots are representative of tissue lysates generated from at least three independent animals for each condition ($n = 3$). Immunoblots were quantified using ImageJ software. Quantitation includes all independent data from the respective experiments and is presented as the mean ± s.e.m. **d**, **e** Statistical analysis was performed using two-way ANOVA (*$p < 0.05$)

although it does present gastrointestinal complications. Sulfonylureas are also established therapies to promote insulin production, but they are associated with risk of hypoglycemia. Thiazolidinediones, such as rosiglitazone, which function as PPAR (peroxisome proliferator-activated receptor) agonists, are associated with weight gain and increased risk of heart failure. More recently, SGLT2 (sodium-glucose transporter 2) inhibitors were shown to promote a decrease in blood glucose due to increased excretion through the kidneys, although there have been indications of increased risk of diabetic ketoacidosis and urinary tract infections[1,31]. Overall, there are several treatment options currently available, but they are associated with significant side effects and the incidence of diabetes and obesity continues to rise. Consequently, novel therapeutic approaches are required.

The effects of insulin and leptin are coordinated for control of whole-body glucose homeostasis and energy balance. Insulin,

produced by the pancreas, regulates glucose uptake into muscle and other tissues, through activation of specific transporters, with the effect of controlling glucose levels in the blood[32]. In addition, it promotes glucose storage in the liver as glycogen, as well as promoting fatty acid synthesis. Leptin, which is produced by adipose tissue, is a major regulator of food intake and energy expenditure[30,33]. In addition, it contributes to maintaining glucose homeostasis[30]. Insulin exerts its effects by binding to the insulin receptor protein tyrosine kinase and initiating signaling pathways, in particular, those leading to phosphorylation and activation of AKT[34]. Leptin also functions through tyrosine phosphorylation-dependent signaling, in this case through activation of a receptor-associated kinase, JAK2[26]. Diabetes and obesity are diseases of insulin and leptin resistance, in which the signaling responses to the hormones are attenuated. Therefore, an ideal therapeutic target would be one that serves as a negative regulator of both signaling pathways, inhibition of which could

restore a normal hormonal response. PTP1B is one such target that counters the tyrosine phosphorylation events that promote insulin and leptin signaling and may address simultaneously issues of excess weight and disrupted glycemic control.

Gene targeting studies demonstrated that PTP1B-null mice are healthy, display enhanced insulin sensitivity, do not develop type 2 diabetes and are resistant to obesity when fed a high fat diet[6]. Consequently, many programs were established in the pharmaceutical industry to generate small molecule inhibitors of PTP1B. Although potent, specific and reversible active site-directed inhibitors of PTP1B were developed, their high affinity depends upon charge, which limits their bioavailability and drug development potential. Consequently, industry views PTPs as challenging targets. Innovative approaches are required to generate inhibitors of this highly validated target that exhibit greater drug development potential. To try to meet this challenge, we have adopted a novel strategy that exploits a physiological mechanism for regulation of PTP function. Previously, we and others demonstrated that PTP1B is transiently oxidized in response to insulin signaling, and this modification is important for the optimal activation of the insulin receptor tyrosine kinase[16,17]. We found that reversible oxidation induced profound conformational changes in PTP1B, which present new binding surfaces that are unique to the reversibly oxidized form of the protein. We exploited this to generate recombinant antibodies, such as the intrabody scFv45, to trap PTP1B-OX in the catalytically inactive form[17]. We have demonstrated that sequestering PTP1B in the oxidized, inactive state can promote insulin signaling in cells. In this study, we have now demonstrated redox regulation of PTP1B function also in response to leptin signaling. Furthermore, expression of scFv45 led to enhanced leptin signaling in a hepatic stellate cell model. These data suggest that a small molecule scFv45-mimetic that can bind and stabilize PTP1B-OX in its inactive state may enhance both insulin and leptin sensitivity, and thus may underlie a viable therapeutic approach to treatment of diabetes and obesity.

By applying several complementary strategies, we have generated an understanding at a molecular level of the interaction between scFv45 and PTP1B-OX. Previous drug development efforts targeting the active site of PTP1B were hampered by cross reactivity with the closely related paralogue TCPTP. In contrast to PTP1B-null mice, homozygous deletion of TCPTP is lethal due to a systemic inflammatory disease[35]. Therefore, despite the ability of TCPTP also to regulate insulin and leptin signaling[36,37], drug discovery efforts have focused on inhibitors that are specific for PTP1B over TCPTP. A striking feature of scFv45 is its specificity for PTP1B-OX; it does not bind to TCPTP in either oxidized or reduced states[17]. Our analysis demonstrates that a cluster of basic residues unique to PTP1B over TCPTP were required for interaction with acidic residues in the complementarity-determining regions of those scFv constructs that bound to the oxidized form of the phosphatase. Our small molecule screen identified sanguinarine, and ultimately chelerythrine, as inhibitors that functioned also by stabilizing PTP1B-OX; these small molecules demonstrated similar specificity to scFv45 and appear to interact with an overlapping site in the phosphatase.

Chelerythrine displayed remarkable specificity in its interaction with PTP1B. In assays in vitro, it was selective for stabilization of the oxidized form of the enzyme, but did not inhibit the reduced, active form of PTP1B. These effects were seen with both TCEP (thiol-free) and DTT (thiol-dependent), suggesting that its effects were not exerted indirectly through compromising the function of the reducing agent. Furthermore, mutation of basic residues in PTP1B, to generate PTP1B-mut2 (Fig. 1a), produced a form of the phosphatase that was insensitive to chelerythrine (Fig. 5c), again pointing to a direct effect on the enzyme rather than a non-

specific effect, such as aggregation. Finally, the fact that, unlike chelerythrine, the closely related molecule protopine did not stabilize PTP1B-OX further reinforces the specificity of this association. Nevertheless, chelerythrine has been reported to act on other targets in the cell. Most notably, there have been several reports that it is an inhibitor of PKC[38]; however, more recently, this has been called into question[39,40]. Furthermore, the related molecule sanguinarine, which was the initial hit in our LOPAC library screen, is not an inhibitor of PKC[41]. In our cell-based studies, the effects of chelerythrine on insulin signaling in 293T cells were abrogated by expression of catalase. Similarly, its effects on leptin signaling in hepatic stellate cells were attenuated by the NOX-antagonist apocynin (Fig. 7e), indicative of the importance of ROS production for the effects of the compound. Consistent with this mechanism of action, antioxidants were reported to block the effects of chelerythrine on shedding of heparin-binding EGF[42] and the activation of JNK and p38[43]. Therefore this work suggests a new target and mechanism of action for the compound, especially in the context of insulin and leptin resistance.

Consistent with the demonstration that obesity is associated with reversible PTP oxidation in the liver[44], our study revealed that PTP1B was reversibly oxidized in response to insulin and leptin administration to animals fed a chow diet and that the levels of PTP1B-OX were elevated in the high fat diet-fed mouse model of obesity (Fig. 8a, b). In addition to its effects in cells, our studies demonstrated that chelerythrine promoted signaling in response to insulin and leptin, enhanced glucose homeostasis and promoted weight loss in high fat diet-fed mice (Fig. 9). Overall, our data are consistent with a model in which chelerythrine stabilizes the oxidized form of PTP1B in the obese animals, thereby promoting whole-body glucose homeostasis. In addition to targeting PTP1B-OX specifically over TCPTP, it is important to note that this approach will only target the pool of PTP1B that is susceptible to reversible oxidation and inactivation. Presumably, because this is the pool of the phosphatase that has been selected for acute regulation in response to hormone, this is also the critical pool of PTP1B that is important for regulation of downstream signaling. Consequently, this strategy may decrease any side effects that would result from targeting the total pool of PTP1B.

Overall, our study illustrates a strategy to target therapeutically a major signaling enzyme that is considered by the pharmaceutical industry to be too difficult to inhibit with active site-directed small molecules. Our intrabody-guided development of a functional high-throughput screening assay provides proof-of-concept in identifying small molecule mimetics as an approach to early stage drug discovery. The structural attributes of scFv45 binding can be further applied to refine future drug development efforts through analysis of CDR-mimetic pharmacophores. It is important to note that although chelerythrine itself is unlikely to be a drug candidate, we hope that this platform may serve as a starting point for the development of novel, more potent and drug-like molecules with which to address this major unmet medical need.

## Methods

**Reagents**. All common reagents were obtained from Thermo Fisher Scientific (Waltham, MA) or Sigma-Aldrich (St. Louis, MO). Difluoro-4-methylumbelliferyl phosphate (DiFMUP) was obtained from Invitrogen (cat # D22065). The solid black 384-well plates (cat# 3573) for the compound screening assay were from Corning Incorporated (Corning, NY). The LOPAC compound Library was obtained from Sigma-Aldrich (St. Louis, MO).

**Antibodies**. The antibodies used in the study: mouse monoclonal anti-phosphotyrosine antibody (catalog 05–321, clone 4G10; Millipore); anti-pY1162/1163-IR-β (catalog 700393, clone 97H9L7; Invitrogen); anti-IR-β (catalog sc-711, clone C711; Santa Cruz Biotechnology Inc.); anti-actin (catalog A2228, clone AC-

74, Sigma-Aldrich); anti-p-T308 AKT (catalog 13038, clone D25E6; Cell Signaling Tech), anti-AKT (catalog 4691, clone C67E7; Cell Signaling Tech), anti-JAK2 (catalog 3230, clone D2E12; Cell Signaling Tech), anti-catalase (catalog ab16731; abcam), anti-HA (catalog 11583816001, clone 12CA5; Roche), and anti-PTP1B (EP1841Y; abcam). Antibodies were used at dilutions recommended by the manufacturers.

**Site-directed mutagenesis.** Site-directed mutagenesis was performed using QuikChange multisite-directed mutagenesis kit (Stratagene) as per manufacturers protocol. PTP1B (1–321) and TCPTP (1–317) cloned in pET21b vector were used as templates to generate -mut1 and -mut2 mutant plasmids, respectively. The PTP1B-mut1 construct was generated using TTACCAGGATATCCGA-CATGAAGCCGACTATCCACATAGAGAGGCCAAGCTTCCTAAGAAC as forward and GTTCTTAGGAAGCTTGGCCACTCTATGTGGA-TAGTCATGGGCTTCATGTCGGATATCCTGGTAA as reverse primers. PTP1B-mut2 construct was generated using CCATGTAGAGTGGCCAAGTTTCCTGA-GAACAGAAACCAAATAGGTACAG as forward and CTGTACC-TATTTCGGTTTCTGTTCTCAGGAAACTTGGCCACTCTACATGG as reverse primers. The TCPTP-mut1 construct was generated using GTACTTG-GAAATTCGAAATGAGTCCAGTGACTTTCCTTGTA-GAGTGGCCAAGTTTCCAGAAAC as forward and GTTTTCTGGAAACTTGGCCACTCTA-CAAGGAAAGTCACTGGACTCATTTCGAATTTCCAAGTAC as reverse primers. TCPTP-mut2 construct was generated using TATCCTCATAGAGTGGCCAAGTTACCAAAAAACAAAAATCGAAACAGA-TACAGAGAGATG as forward and CATCTCTGTATCTGTTTC-GATTTTTGTTTTTTGGTAACTTGGCCACTCTATGAGGATA as reverse primers.

**Purification of wild-type and mutant PTP1B and TCPTP.** Wild-type and mutant forms of PTP1B and TCPTP containing a 6xHis-tag were expressed in BL21 (DE3)-RIL *Escherichia coli* using LB. Cells were resuspended in lysis buffer (50 mM Hepes pH 7.2, 150 mM NaCl, 10 mM imidazole, 2 mM TCEP) containing cOmplete EDTA-free Protease Inhibitor Cocktail (Roche), and then lysed using sonication at 4 °C. Lysates were clarified by centrifugation, with subsequent protein purification performed by gravity flow using Ni-NTA (Qiagen) column chromatography. Protein was further purified using an AKTA FPLC connected to a HiLoad 16/600 Superdex 75 pg size-exclusion chromatography column (GE Healthcare Life Sciences) into 50 mM HEPES pH 7.4, 100 mM NaCl, and 2 mM dithiothreitol. Protein was used immediately or stored at −80 °C in in 50 mM HEPES pH 7.4, 100 mM NaCl, 2 mM dithiothreitol, and 25% glycerol.

**PTP1B oxidation and reactivation.** Purified PTP1B (1–321) (100 nM) was mixed with increasing concentrations of $H_2O_2$ (0.05–100 mM) in phosphatase assay buffer (50 mM HEPES, pH 7.0, 100 mM NaCl, 0.1% BSA) at room temperature for 5 min. From this reaction mixture PTP1B (10 nM) was used to monitor phosphatase activity using DiFMUP as the substrate, with or without 5 mM TCEP. Activity of each sample was compared to that of the untreated PTP1B in the presence of 5 mM TCEP.

**Binding assay.** To demonstrate interaction between wild-type (wt) and mutant forms of PTP1B-OX or TCPTP-OX and scFv45, we performed an in vitro binding assay using purified recombinant proteins. Purified proteins (wt or mutant form of PTP1B or TCPTP (50 nM)) were reversibly oxidized with 250 mM $H_2O_2$ followed by a buffer exchange using 0.5 mL Zeba spin desalting columns 7K MWCO (Thermo Fisher Scientific) to remove $H_2O_2$. Increasing concentration of purified HA-tagged scFv45 (0–10 μM) was incubated with oxidized proteins in binding buffer (20 mM HEPES, pH 7.4, 300 mM NaCl, 0.05% BSA, 0.05% Tween-20) for 2 h at 4 °C. Anti-HA agarose (Roche, 50% slurry equilibrated in the binding buffer) was added and incubated for 1 h at 4 °C. Protein complexes bound to anti-HA beads were immunoprecipitated and washed three times; first with lysis buffer followed by two more washes with wash buffer (25 mM HEPES, 100 mM NaCl pH 7.4, 0.05% BSA, 0.05% Tween-20, and protease inhibitors). The complexes were separated by SDS-PAGE and analyzed by immunoblotting.

**Analytical size-exclusion chromatography.** Recombinant PTP1B-CASA, PTP1B-CASA-mut2, and scFv45 were purified as described. Proteins were exchanged into freshly prepared 50 mM ammonium bicarbonate pH 8 (Fig. 2a; Supplementary Fig. 1) using 0.5 mL Zeba spin desalting columns 7K MWCO (Thermo Fisher Scientific) as per manufacturers protocol. Designated protein constructs or buffer equivalents were mixed at 1.1 or 1.4 molar excess (excess reciprocated for both PTP1B and scFv45) and incubated on ice 20 min prior to injection (20 μL injection loop; 4–10 μg total protein), separation, and detection on a Dionex Ultimate 3000 HPLC connected to an XBridge Protein BEH SEC 200 Å, 3.5 μm, 7.8 mm × 300 mm column (Waters) flowing 200 mM ammonium acetate pH 7.0 isocratically at 0.86 mL min$^{-1}$ and 25 °C, monitoring absorbance at $\lambda$ 280 nm (Fig. 2a; Supplementary Fig. 1). Injections were performed intermittently with BEH200Å SEC Protein Standard Mix (amount corrected for 20 μL injection loop; Waters Corp.) to monitor peak resolution and retention time consistency among runs.

**Size-exclusion chromatography.** Recombinant PTP1B-CASA, TCPTP-CASA, TCPTP-CASA-mut2, and scFv45 were purified as described. Proteins were exchanged into freshly prepared 50 mM HEPES pH 7.4, 100 mM NaCl, 2 mM DTT, 20% glycerol (Supplementary Fig. 2) using 0.5 mL Zeba spin desalting columns 7K MWCO (Thermo Fisher Scientific) as per manufacturers protocol. Designated protein constructs or buffer equivalents were mixed at 1.1 molar excess of scFv45 to TCPTP and incubated on ice 20 min prior to injection (200 μL injection loop; 200–420 μg total protein), separation, and detection on an AKTA FPLC connected to a HiLoad 16/600 Superdex 75 pg SEC column (GE Healthcare Life Sciences) flowing 10 mM HEPES pH 7.4, 100 mM NaCl isocratically at 1 mL min$^{-1}$ and 4 °C, monitoring absorbance at $\lambda$ 280 nm, collecting 1 mL fractions (Supplementary Fig. 2). Designated protein fractions were mixed 4:1 with reducing 6× SDS loading dye, resolved by SDS-PAGE, and proteins visualized by Imperial Protein Stain coomassie dye R-250 (Thermo Fisher Scientific). Quantification of protein staining was performed with ImageJ software, with protein staining of target fractions being normalized to protein staining of internal loading controls (0.2 μg; lanes 2 and 9) in relation to staining of target protein standard curves (0.2–1.6 μg; Supplementary Fig. 2b, c).

**Protease protection assay.** Recombinant PTP1B-CASA, PTP1B-CASA-mut2, and scFv45 were exchanged into freshly prepared 50 mM ammonium bicarbonate pH 8 as described above. Designated protein constructs were mixed at a 1.1-fold molar excess of scFv45 or buffer equivalent (0.3 μg μL$^{-1}$ PTP1B or 0.24 μg μL$^{-1}$ scFv45, respectively) and incubated as above prior to addition of AspN or LysC endoprotease at 1:500 (w/w) or 1:750 (w/w), respectively. Digests proceeded at 25 °C with aliquots being taken at designated time points (0, 15, 30, 60, 120, 240, and 480 min) and quenched by addition of Complete protease inhibitor cocktail with EDTA (Roche) to a final concentration of 5×. Quenched protein digests were mixed 1:2 with reducing 3× SDS loading dye, resolved by SDS-PAGE, and visualized as described above (Supplementary Fig. 5). Alternatively, quenched protein digests were mixed 1:1 with either 50 mM ammonium bicarbonate pH 8 (for PTP1B: scFv45 complex digests) or with the corresponding digest of scFv45-alone (for digests of PTP1B-alone) to equate both total buffer salt and total protein content across all samples. Samples were then mixed 1:4 with 65% acetonitrile (ACN), 0.1% trifluoroacetic acid (TFA) and spotted 1:1 (at 1 μL) with 10 mg mL$^{-1}$ recrystallized α-cyano-4-hydroxy-cinnamic acid matrix (Thermo Fisher Scientific) dissolved in 65% ACN, 0.1% TFA onto a stainless steel target by the dried-droplet method. Above samples mixed 1:4 with 65% ACN, 0.1% TFA from LysC digestion, were further mixed 4:1 with an approximate average of 212.5 fmol μL$^{-1}$ internal peptide mass standard calibration mixture 2 (AB Sciex) dissolved in 65% ACN, 0.1% TFA, and spotted onto the target plate as described above. To eliminate the inherent bias in spectral acquisition among sample spots due to manual targeting of optimal co-crystals, peptide mass fingerprints of protein digests were analyzed by matrix assisted laser desorption ionization-time of flight mass spectrometry (MALDI-TOF MS) in positive ion mode using either the automated linear or automated reflector mode on an Applied Biosystems Voyager DE-Pro Mass Spectrometer. For each instrument mode and sample condition, the attenuated laser intensity (ranging 1700–1810), extraction delay time (ranging 120–250 nsec), and number of laser shots per spectrum (ranging 250–500), were manually optimized and remained constant throughout each comparable analysis, as were all other instrumental parameters. Each spot was analyzed randomly with a center-bias to ensure that acquired spectra (5–10 per sample spot) were measured from within the droplet boundaries of formed co-crystals, culminating to 140–280 spectral replicates per proteolytic condition (AspN or LysC). Each acquired spectra was individually curated and selected for display based on near average absolute and relative ion intensities, optimal signal-to-noise ratios, and equivalency of these parameters among conditions to be compared. Spectra were baseline corrected and mass calibrated with internal peptides of known identities and masses using the Data Explorer software (Applied Biosystems) (Fig. 2b, c; Supplementary Figs. 3a and 4a). Mass spectral measurements and processing, as outlined above, were obtained from at least two independent experiments for each proteolytic condition. Tandem mass spectrometry (MS/MS) of target ions was performed manually using post-source decay, with spectra being acquired in 10–15 segments of mirror ratios ranging from 0.1 to 1.0. Segments were compiled into single spectra, baseline corrected, and noise filtered using Data Explorer software (Applied Biosystems) and compared to in silico fragmentation using the Protein Prospector software (http://prospector.ucsf.edu/) (Supplementary Figs. 3b and 4b).

**Expression and purification of the intrabody scFv45.** Recombinant scFv45 was purified similar to previously described methods[17]. Briefly, 6xHis-tagged scFv45 was expressed in BL21 (DE3)-RIL *Escherichia coli* using LB. Cells were resuspended in lysis buffer (50 mM Tris pH 7.5, 250 mM NaCl, 10 mM imidazole) containing cOmplete EDTA-free Protease Inhibitor Cocktail (Roche), then lysed by sonication at 4 °C. Lysates were clarified by centrifugation, with initial protein purification being performed by gravity flow using Ni-NTA column chromatography. The 6xHis-tag of scFv45 was removed by proteolysis using recombinant 6xHis-tagged rTEV protease overnight at 4 °C. Uncleaved scFv45-6xHis, cleaved 6xHis-tag, and the 6xHis-tagged rTEV were all separated from cleaved, untagged scFv45 by gravity flow using Ni-NTA column chromatography as above. Protein was further purified

using an AKTA FPLC connected to a HiLoad 16/600 Superdex 75 pg size-exclusion chromatography column (GE Healthcare Life Sciences) in 50 mM HEPES pH 7.4, 100 mM NaCl, and 2 mM dithiothreitol as above. Protein was used immediately or stored at −80 °C.

**Structure determination and refinement**. Initial crystal screening was performed using soluble screen at the High-throughput Crystallization Screening lab (HTSlab) at Hauptman-Woodward Medical Research Institute[45]. Crystals were obtained at 4 °C by the sitting drop vapor-diffusion method. Two hundred nl of scFv45 (13 mg ml$^{-1}$) was mixed with 200 nl of crystallization buffer (0.1 M CAPS pH 10.0, 0.1 M Ammonium Chloride, 40% PEG 400) and placed over a reservoir of 300 μl in volume. scFv45 protein crystals reached a size of $80 \times 20$ microns within a week. Crystals were flash-frozen in liquid nitrogen directly from the crystal drop. X-ray diffraction data were collected at 100 K at Keck Structural Biology Laboratory using Rigaku MicroMax-007HF at a wavelength of 1.54178 Å. Data were processed using XDS[46]. The structure of scFv45 was solved by molecular replacement with Phaser[47] using chicken scFv structure, (PDB ID: 4P49)[48] as a search model, with one molecule in the asymmetric unit. Initially, model building used Autobuild[47], followed by manual model building using Coot[49]. Structure refinement was performed with Phenix.refine[47] using the individual B-factors and occupancies. Water molecules were added during Autobuild[47] and manually using Coot[49]. Ramachandran-plot analysis, carried out with MolProbity[50], showed that 97.30% of the residues were in favored regions; 100% were in allowed regions. For representative numbering of scFv45 residues, the first amino acid denotes the residue directly following the OmpA cleavage site.

**LOPAC screen**. To screen the LOPAC library against PTP1B-OX, 0.5 μl of 1 mM compound in 100% DMSO was added to 384-well plates, resulting in a final compound concentration of 10 μM for the primary screen. Recombinant PTP1B (100 nM) was reversibly oxidized and inactivated with $H_2O_2$ (4 mM). The phosphatase activity was restored by a quick buffer exchange and addition of reducing agent (TCEP). PTP1B-OX was incubated with LOPAC compounds (10 μM) in the assay buffer (50 mM HEPES pH 7.0, 100 mM NaCl, 0.005% Tween, 2 mM EDTA) and the effect of individual compound on stabilizing the reversibly oxidized conformation was assessed by the phosphatase assay under reducing conditions using DiFMUP as substrate. scFv45 was used as positive control. Z′-factor values were always between 0.6 and 0.8 indicating that the assay is robust.

**Isothermal titration calorimetry**. The studies were performed using a VP-ITC system from GE Healthcare. All titrations were performed at 25 °C in de-gassed buffer (50 mM HEPES pH 7.0, 100 mM NaCl, 2 % DMSO, and 1 mM TCEP). The protein concentrations of PTP1B-CASA and PTP1B-CASA-mut2 were each 50 μM. Protein samples used in ITC were dialyzed completely against the buffer. Inhibitor (10 mM) in DMSO was diluted to make a stock solution of 0.2 mM in de-gassed buffer, which was then used as titrant in the syringe. The instrument was calibrated by using the heat of dilution of NaCl in water. By titrating buffer into protein solution the heat generated due to protein dilution was estimated and was found to be negligible. The heat of ligand dilution was corrected by subtracting the average heat of injection after saturation. Origin software was used to fit and analyze the data.

The data in Supplementary Fig. 9 are representative of three independent experiments for PTP1B-CASA, in which the affinities were measured as $3.25 \pm 0.36$, $2.83 \pm 0.15$, and $3.44 \pm 0.23$ micromolar, and two independent experiments for PTP1B-CASA-mut2, in which we did not detect saturable binding.

**Molecular modeling**. Protein and Ligand Preparation: Within Schrödinger's Maestro interface, the Protein Prep Wizard was applied to process the following pdb structures: 1OEM (PTP1B-OX), 1A5Y (PTP1B, closed form), and 1T4J (PTP1B, open form). The enzymes were treated to correct any mismatched bonds, add missing hydrogens, remove bulk water, and assign atom types, bond orders and atomic partial charges. In addition, a simple minimization was performed in this step using the OPLS2005 force field to a default RMSD of 0.3 Å. Levamisole, sanguinarine, and chelerythrine were minimized (50 iterations per input structure, SD method), and a set of low-energy conformations were obtained for each using the LigPrep tool and the OPLS2005 force field. LigPrep generates atom types and other parameters specific for the Glide module; in addition, alternate tautomers, ionization states, and ring geometries are sampled to broaden the sets of ligand input conformations for docking.

SiteMap: Site map begins with an initial search that identifies potential superficial small molecule binding sites on the enzyme by using a grid of "site points." Next, contour maps are produced that include donor, acceptor, and hydrophobic pocket regions within the sites. A SiteScore is obtained that incorporates some of the following properties: "the size of the site; the degrees of enclosure by the protein and exposure to solvent; the tightness with which the site points interact with the receptor; the hydrophobic and hydrophilic character of the site and the balance between them; the degree to which a ligand might donate or accept hydrogen bonds." (Ref: Schrödinger 2016-1 SiteMap User Manual).

The SiteScore is defined as follows:

$$\text{SiteScore} = 0.0733 \sqrt{(n)} + 0.6688\, e^{-0.20\, p}$$

where, $n$ = number of site points (100 max), $e$ is the enclosure score, and $p$ is the hydrophilic score. A SiteScore >1 is considered significant and worthy of consideration as a potential binding site (Ref: Schrödinger 2016-1 SiteMap User Manual). Five top-scoring sites were identified for 1OEM, 1A5Y, and 1T4J, respectively.

Glide docking: The docked models of 1OEM and levamisole, sanguinarine, and chelerythrine were generated with the Glide module. In the initial step of Glide docking, a grid was constructed in the receptor, defining the binding site and accounting for potential Van der Waals, polar, and charged interactions. For 1OEM, the centroid of site 5 was defined as the "ligand" for grid generation with a default "inner box" of 10 Å$^3$ and an "outer box" of 26 Å$^3$. Levamisole, sanguinarine, and chelerythrine were docked flexibly within the grid in SP mode for initial model development.

Poses were ranked with the GlideScore function:

$$\text{GScore} = 0.065 * \text{vDW} + 0.130 * \text{Coul} + \text{Lipo} + \text{HBond} + \text{Metal} + \text{BuryP} + \text{RotB} + \text{Site}.$$

Restrained minimization: A restrained minimization was performed for chelerythrine to identify whether hydrogen bond contacts may be made between the methoxy oxygen atoms of chelerythrine and amine atoms of key residues K41, N44, and R45. These residues do not appear to be involved in structural stability or to impact the conformation of the enzyme. The distances between the proposed interacting atoms was set at 4 Å, and after 25,000 iterations using the PRCG method, set to converge on movement at 0.005 convergence threshold, the minimization concluded successfully. Chelerythrine and Y46 were held fixed during the calculation. No severe distortions were observed in the enzyme structure.

Software references: Schrödinger's 2015–3 and 2016–7 Small Molecule Drug Discovery Suite: Maestro v10.3, v10.9; Macromodel v10.7, v11.1; Glide v6.8, v70014; SiteMap v3.4.013. Schrödinger, LLC: New York, NY, 2015, 2016.

**Displacement assays**. To test whether there was overlap in the binding site for chelerythrine and scFv antibodies that stabilized PTP1B-OX, we performed an in vitro competition binding assay using purified components. The antibody scFv45 displays nanomolar affinity for PTP1B-OX; therefore, we characterized additional scFvs to identify antibodies with micromolar affinities for PTP1B-OX, closer to that of chelerythrine. We focused on scFv106 and scFv67. Purified PTP1B (50 nM) was reversibly oxidized with 250 μM $H_2O_2$ followed by a buffer exchange to remove excess $H_2O_2$. Increasing concentration of purified scFv antibody (0–10 μM) was incubated with oxidized proteins in binding buffer (20 mM HEPES pH 7.4, 300 mM NaCl, 0.05% BSA, 0.05% Tween-20) for 2 h at 4 °C. Anti-HA agarose (Roche, 50% slurry equilibrated in the binding buffer) was added and incubated for 1 h at 4 °C. Protein complexes bound to anti-HA beads were precipitated and washed three times; first with lysis buffer followed by two more washes with wash buffer (25 mM HEPES pH 7.4, 100 mM NaCl, 0.05% BSA, 0.05% Tween-20, and protease inhibitors). The protein complex was incubated with varying concentrations of chelerythrine (0–20 μM) for 2 h at 4 °C, following which the beads were washed as described above. The complexes were resolved by SDS-PAGE and analyzed by immunoblotting.

**In vitro substrate trapping of PTP1B by Trx1 mutant variants**. Thioredoxin (TRX), which functions to reduce oxidized cysteines in target proteins, contains an essential CXXC motif, in which the first cysteine forms a mixed disulfide intermediate with an oxidized cysteine in the target protein. The second cysteine then serves a "resolving" function—it forms a disulfide bond with the first cysteine in the CXXC motif, with the resulting transfer of electrons leading to reduction of the target protein. We utilized an assay using a "trapping mutant" form of thioredoxin in which the second cysteine is mutated to serine, which stabilizes the mixed disulfide intermediate with the target substrate.

Recombinant variants of N-terminally His-tagged human TRX1 (0.1 mg, WT, and trapping mutant, C35S) were reduced with 20 mM DTT in 1 mL PBS on ice for 30 min, then desalted on a Sephadex PD-10 column. Ten μg each of TRX1-WT and TRX1-C35S proteins was used in the trapping experiment. PTP1B (40 μg ml$^{-1}$) was incubated with $H_2O_2$ (0, 0.1 and 0.5 mM) for 10 min at room temperature and excess $H_2O_2$ was removed by desalting. Reduced and oxidized forms of PTP1B (20 μg) were incubated with WT or trapping mutant TRX1 for 20 min at room temperature in the absence and presence of chelerythrine (5 μM) in 1 mL PBS. Ni-NTA beads were added and incubated for 1 h at 4 °C. Protein complexes bound to Ni-NTA beads were washed three times with PBS (pH 7.4) containing 40 mM Imidazole, 0.05% BSA, 0.05% Tween-20, and protease inhibitors. The complexes were resolved by SDS-PAGE and blotted for PTP1B (FG6) and TRX1 (sc-58439, Santa Cruz Biotechnology).

**Insulin signaling in HEK 293T cells**. To study the effect of chelerythrine and protopine on insulin signaling we used 293T cells (ATCC) and cultured in DMEM containing 10% FBS. For insulin signaling, cells were serum starved for 16 h and then treated with chelerythrine (2 μM) or protopine (2 μM) for 1 h. Following which, cells were treated with 25 nM insulin for various times (0, 5, 10, 20 min) at 37 °C. Cells were washed with PBS and lysed in de-gassed RIPA lysis buffer (25 mM HEPES pH 7.5, 150 mM NaCl, 0.25% Deoxycholate, 10% Glycerol, 25 mM NaF, 10 mM $MgCl_2$, 1 mM EDTA, 1% Triton X-100, 0.5 mM PMSF, 10 mM Benzamidine, cOmplete protease inhibitor cocktail (Roche)). Soluble proteins were harvested by centrifugation at 13,000×$g$ for 10 min at 4 °C and quantitated. Total proteins in the cell lysates were separated by SDS-PAGE and tyrosyl phosphorylation of the IR-β subunit and downstream AKT activation was monitored. To test the effect of suppressing $H_2O_2$ levels on the function of the compounds, catalase was ectopically expressed in cells for 24 h. Following which, the cells were treated with the compounds as mentioned above. Phosphorylation of the AKT activation loop at residue threonine 308 (T308) in response to insulin stimulation was observed with the phospho-specific antibody. Catalase expression was detected in the cell samples with a rabbit polyclonal antibody (Abcam, ab52477).

**Effects of chelerythrine after PTP1B variant overexpression**. Mammalian expression plasmids used in this study were as follows: pCMV-FLAG and pCMV-FLAG-PTP1B (wt and -mut2). Empty vector, FLAG-PTP1B-wt, or FLAG-PTP1B-mut2 were transfected into 293T cells for 24 h, following which, cells were serum starved for 8 h and then treated with chelerythrine (2 μM) for 1 h. Subsequently, cells were treated with 10 nM insulin for various times (0, 10, 20 min) at 37 °C. Cells were washed with PBS and lysed in de-gassed RIPA lysis buffer (25 mM HEPES pH 7.5, 150 mM NaCl, 0.25% Deoxycholate, 10% Glycerol, 25 mM NaF, 10 mM $MgCl_2$, 1 mM EDTA, 1% Triton X-100, 0.5 mM PMSF, 10 mM Benzamidine, cOmplete protease inhibitor cocktail (Roche)). Soluble proteins were harvested by centrifugation at 13,000×$g$ for 10 min at 4 °C and quantitated. Total proteins in the cell lysates were separated by SDS-PAGE and tyrosyl phosphorylation of the IR-β subunit was analyzed by immunoblotting.

**Leptin signaling in hepatic stellate cells**. Human hepatic stellate cells (HSteCs; ScienCell research laboratories, CA) were cultured according to manufacturer's instructions. Cells were seeded on poly-L-lysine coated dishes and maintained in Stellate Cell Medium (SteCM, Cat. #5301). When cells reached 70% confluency they were split for various experiments. To study leptin signaling, HSCs were incubated in serum-free DMEM for 24 h and then stimulated with varying concentration of leptin (0, 10, 25, 50, 100, and 200 ng ml$^{-1}$) for 15 min. To study the effect of chelerythrine and protopine, serum starved cells were incubated with the compounds at varying concentrations (0–5 μM) for 1 h prior to leptin stimulation. In some groups, cells were pre-incubated (60 min) with the NADPH oxidase inhibitor apocynin (100 nM), before stimulation with leptin.

**Measurement of intracellular ROS**. Cells cultured in 24-well plates were washed with PBS and changed to serum-free media 24 h before ROS measurement. Cells were incubated with 2′,7′-dichlorofluorescin diacetate (DCFDA) (8 μM) for 20 min at 37 °C. Cells were then rinsed twice with serum-free medium and stimulated with varying concentrations of leptin. DCFDA fluorescence was detected at excitation and emission wavelengths of 488 and 520 nm, respectively, using a multi-well fluorescence scanner (Spectramax Gemini, molecular devices) continuously for 30 min.

**Animal experiments**. Animal experiments were performed according to protocols approved by the Institutional Animal Use and Care Committee of Cold Spring Harbor laboratory. Ten week-old male C57bl6/J mice fed chow diet or high fat diet (D12492) were acclimatized for 10 days under standard conditions before experiments. Mice were intraperitoneally (i.p.) injected once daily with vehicle, 3 mg kg$^{-1}$ of chelerythrine or 3 mg kg$^{-1}$ of protopine for 30 days. Mice were killed after 6 h of fasting, and serum samples were collected to measure the metabolic parameters and tissue samples were collected for studying changes in signaling. No animals were excluded from the analysis.

**Measurement of PTP1B oxidation in vivo**. Mice fed normal or high fat diet were administered saline, insulin (1 U kg$^{-1}$) or leptin (1 mg kg$^{-1}$) subcutaneously (s.c.). Within 30 min after the treatment, animals were killed, tissues were dissected, weighed, flash-frozen in liquid nitrogen, and stored at −80 °C. For PTP oxidation, liver tissue or hypothalamus were homogenized under anaerobic conditions in de-gassed, ice-cold lysis buffer (25 mM HEPES pH 7.4, 100 mM NaCl, 10 mM N-ethylmaleimide, 0.25% deoxychloate, 1% Triton X-100, 25 mM NaF, 10 mM $MgCl_2$, 1 mM EDTA, 10% glycerol, and cOmplete protease inhibitor cocktail) for 1 h at 4 °C. Lysates were clarified by centrifugation at 16,000×$g$ for 20 min and diluted 100-fold using 25 mM HEPES, pH 7.4, 100 mM NaCl. Lysates were quantitated and 1 mg of each lysate was incubated with scFv45 (0.1 mg ml$^{-1}$) for 90 min. The protein complexes were precipitated with anti-HA-agarose beads and washed three times; first with lysis buffer followed by two more washes with wash buffer (25 mM HEPES pH 7.4, 100 mM NaCl, 0.05% BSA, 0.05% Tween-20, and protease inhibitors). Complexes were separated by SDS-PAGE and immunoblotted.

**Metabolic measurements**. Glucose in tail blood was measured using a glucometer (One-Touch Basic; Lifescan, CA). For glucose tolerance tests (GTTs), mice were fasted for 10 h and then injected with 20% D-glucose (2 mg g$^{-1}$ body weight) and the blood glucose was monitored immediately before and at 15, 30, 60, and 120 min following the injection. For insulin tolerance tests (ITTs), 4 h fasted animals were given insulin (0.75 mU g$^{-1}$) and blood glucose was measured immediately before and at 30, 60, and 120 min post-injection. Statistical analysis was performed using ANOVA for both GTT and ITT.

**Statistics and reproducibility**. No statistical method was used to predetermine sample sizes. Samples were not randomized. The investigators were not blinded to allocation during experiments or outcome assessment. Sample sizes and statistical tests for each experiment are denoted in the figure legends. Statistical analysis and tests were chosen based on established protocols. Each immunoblot was performed at least three times (biological replicates) for quantification. Statistical analysis was performed by using the two-tailed Student's $t$-test (Figs. 6a, b and 7a, b), paired two-way ANOVA (Figs. 8b and 9b–e). $p$-values can be found in corresponding figure legends.

**Data availability**. Source data for figures are available from the corresponding author upon request. Coordinates and structure factors for scFv45 have been deposited in the Protein Data Bank under accession code 5VF6.

All uncropped blots are provided in Supplementary Fig. 18.

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

## Acknowledgements

We thank P.O. Vacratsis (University of Windsor) for generous use of the MALDI-TOF mass spectrometry and HPLC infrastructure and resources, and Elad Elkayam (Cold Spring Harbor Laboratory) for help with the structural studies. This research was supported by NIH Grant GM55989 to N.K.T., the Cold Spring Harbor Laboratory Women in Science Award to L.J.-T., the CSHL Cancer Centre Support Grant CA45508, and the Robertson Research Fund of Cold Spring Harbor Laboratory. L.J.-T. is an investigator of the Howard Hughes Medical Institute. N.K.T. is also grateful for support from the Don Monti Memorial Research Foundation. C.A.B. is supported by a Natural Sciences and Engineering Research Council of Canada (NSERC) Post-Doctoral Fellowship.

## Author contributions

N.K. carried out the activity assays in Fig. 1, the screen in Fig. 4, the mutational analysis in Fig. 5, and the analysis in cell and animal models of insulin and leptin signaling presented in Figs. 6 and 7. C.A.B. carried out the analytical size-exclusion chromatography and mass spectrometry-based structural proteomics analysis presented in Fig. 2. I.R. carried out the analysis of scFv45 interaction with PTP1B and TCPTP presented in Fig. 1. O.K.S., in collaboration with A.T. and L.J.-T., carried out the structural analysis in Fig. 3. C.M.G. carried out the modeling in Fig. 5. A.H. initiated the project to identify and characterize the conformation-sensor antibody scFv45, and generated many reagents used in this study. N.K.T. directed the study and wrote the paper, with input from all of the authors. All of the authors contributed generally to discussions throughout the project.
