## [Peer Review File · Nature Communications]

Reviewers' comments:

Reviewer #1 (Remarks to the Author):

This is a very nice study demonstrating the therapeutic potential of selectively targeting an oxidized pool of PTP1B with small molecule compounds. Although PTP1B is a highly validated target for diabetes and obesity, traditional drug discovery effort aimed at inhibiting the phosphatase activity of PTP1B has failed to produce any clinical candidates. The work provides solid new information that will be of interest to others in this field. The authors should consider the following comments that might improve the manuscript.

1. To further support the conclusion that chelerythrine and scFv45 share the same overlapping binding site in PTP1B, it would be important to show that chelerythrine blocks scFv45 binding to PTP1B-OX. Does expression of scFv45 abrogate the effect of chelerythrine on insulin signaling?
2. Does chelerythrine have any effect on insulin/leptin signaling in the basal state?
3. It would significantly strengthen the paper if the authors could provide in vivo target engagement data for chelerythrine using thermal shift assays.
4. There is a disconnect between the measured IC₅₀ for chelerythrine (5 uM) and the observed cellular efficacy (<2 uM). Any thought?
5. For all panels (Western blots), there needs to be a more thorough quantitative analysis including determination of statistical significance through replicate experiments.
6. It is not clear whether changes in Fig 7D and E are statistically significant.
7. More than 20 single nucleotide polymorphisms (SNPs) that are associated with increased risk of type 2 diabetes have been identified within the PTPN1 gene---are these SNPs gain-of-function?

Reviewer #2 (Remarks to the Author):

Insulin signaling is critical for glucose homeostasis and the development of diabetes and its complications. The protein tyrosine phosphatase PTP1B occupies a critical position in signal transduction as a negative regulator of insulin and leptin signaling. PTP1B is reversibly oxidized and inactivated to allow the insulin receptor to be activated. Since knockout mice lacking PTP1B are viable and lean with improved glucose tolerance. Developing inhibitors of PTP1B is therefore of great interest to develop drugs against the epidemic of type II diabetes. Developing inhibitors of the active site in PTP1B has met with problems and as an alternative approach Tonks and coworkers have previously generated an antibody (scFv45) that binds to the oxidized inactive PTP1B and prevents its reductive activation. In this study they have characterized the binding mode of this antibody and identified a small molecule inhibitor of the PTP1B with the desired properties to only bind the oxidized PTP1B and not to PCPTP, which is a related phosphatase which can be associated with severe side effects if inhibited. They have also made extensive animal experiments to characterize the metabolic outcome in treated animals. Overall this study deals with a highly important and interesting subject. The manuscript is of highest caliber and contains a wealth of data. The presentation is excellent. The extensive investigation and its results are highly significant and the paper presents a novel paradigm for inhibition of the enzyme PTP1B.

Reviewer #3 (Remarks to the Author):

Krishnan et al. mapped the site of interaction between oxidized PTP1B and scFv45 to a loop comprising residues 36-41 by mutagenesis and independently by structural proteomics. Substitution of three residues in this loop abolished the interaction without significant effects on catalytic activity. In silico docking indicated that acidic residues of ScFv45 have an important role in the interaction and that Lys41 in the interaction loop forms a hydrogen bond with Asp93 of ScFv45. A screen for compounds that lock oxidized PTP1B in its inactive conformation identified two candidate and a derivative, Chelerythrine, had even more pronounced effects. Docking of these compounds suggested a binding site overlapping the binding site of ScFv45 and Chelerythrine inhibited PTP1B-OX 15-fold more than K36AK41A PTP1B-OX. Insulin signaling was enhanced by chelerythrine, which was abolished by catalase treatment. Leptin signaling induced PTP1B oxidation and chelerythrine treatment enhanced downstream Leptin signaling in tissue culture cells. Chelerythrine treatment improved metabolism in mice on a high fat diet. The authors conclude that their results provide proof-of-concept that stabilization of PTP1B in an inactive, oxidized conformation by small molecules can promote insulin and leptin signaling.

This is an interesting paper. In general, PTPs have proven difficult to target by inhibitors and this paper indicates a new mechanism to specifically inhibit PTP1B by locking the oxidized form in its inactive state.

Points:

1. The authors convincingly show that chelerythrine locks oxidized PTP1B in the inactive conformation. Is this interaction irreversible? Does prolonged incubation of chelerythrine-treated oxidized PTP1B with reducing agents lead to reduction and release of chelerythrine? Does chelerythrine somehow block reduction of the catalytic cysteine, e.g. by blocking access of reducing agents?
2. Fig. 6. Chelerythrine has a modest effect on insulin signaling. Given that chelerythrine locks PTP1B in an inactive state, the most profound response will presumably become evident when the transient response to insulin declines at later time-points. Are the effects of chelerythrine more profound at later time points? Note that in the control, the 20 min timepoint shows the greatest response also in the control. Along the same lines, the response to leptin in the presence of ScFv45 (panel D) shows a plateau at 20, 30 and 60 min for the control and the ScFv45 expressing condition, which is surprising, because one would expect higher responses at later time points. The difference between control and scFv45 expressing cells appears modest in panel D, yet quantification shows a 7-fold increase. This quantification appears to overestimate the difference.
3. Discussion. Chelerythrine has been reported to act on other targets in cells, particularly PKC. Are the earlier described effects of chelerythrine on cells consistent with PTP1B inhibition?

Minor points:

1. Fig. 3A. The numbers of the indicated residues in the Figure and the legend are not consistent.
2. Fig. 5D. Reduced and oxidized PTP1B are represented in different colors, gray and blue. Please indicate in the legend which is which.
3. Fig. 6D-G. For clarity, the authors should mention "leptin" as the factor that was added to the cells.
3. Fig. 7. The legends of panel A and B are switched. Legend to panel A indicates that oxidation of PTP1B was measured. How was this measured?
4. Supplementary Fig. 9. Indicate highlighted residue names and numbers in panel B.

Response to Reviewers' comments:

Overall, we are grateful to the referees for their supportive and constructive comments. Our responses to the various points that were raised are highlighted in blue.

Reviewer #1 (Remarks to the Author):

This is a very nice study demonstrating the therapeutic potential of selectively targeting an oxidized pool of PTP1B with small molecule compounds. Although PTP1B is a highly validated target for diabetes and obesity, traditional drug discovery effort aimed at inhibiting the phosphatase activity of PTP1B has failed to produce any clinical candidates. The work provides solid new information that will be of interest to others in this field. The authors should consider the following comments that might improve the manuscript.

1. To further support the conclusion that chelerythrine and scFv45 share the same overlapping binding site in PTP1B, it would be important to show that chelerythrine blocks scFv45 binding to PTP1B-OX. Does expression of scFv45 abrogate the effect of chelerythrine on insulin signaling?

The problem with such an experiment is the relative affinities of chelerythrine and scFv45 for PTP1B-OX, micromolar versus nanomolar, which represent an insurmountable obstacle to a competition experiment. In an attempt to address this point, we characterized some additional scFvs to identify antibodies with affinities for PTP1B-OX closer to that of chelerythrine. The results with two such molecules, scFv67 and scFv106, are presented in a new Supplementary Figure 10. Like scFv45, these antibodies recognise PTP1B-OX, but not PTP1B-mut2-OX, suggesting a similar binding mode. Particularly for scFv106, we were able to demonstrate that chelerythrine competed with the antibody for binding to PTP1B-OX.

2. Does chelerythrine have any effect on insulin/leptin signaling in the basal state?

As shown throughout Figure 6, there is no impact of chelerythrine on basal signaling in the absence of insulin or leptin. Consistent with the effects of scFv45, this suggests a function as an insulin-sensitizer, rather than as an insulin-mimetic. This is consistent with the model that the target of chelerythrine, PTP1B-OX, is only generated following stimulation with insulin or leptin.

3. It would significantly strengthen the paper if the authors could provide in vivo target engagement data for chelerythrine using thermal shift assays.

Unfortunately, it was not possible to include such data. There was insufficient time both to set up the technology in the lab and to respond to the referee's comments in a timely manner. In addition, I tried to engage specialist help – I contacted Matthew Hall at the National Center for Advancing Translational Sciences to request assistance with this, but it has not been possible to conduct the assays in that setting.

4. There is a disconnect between the measured IC50 for chelerythrine (5 uM) and the observed cellular efficacy (<2 uM). Any thought?

We present several lines of data that support PTP1B-OX as the target of chelerythrine in vivo, despite this apparent difference in IC50 and cellular potency. The reason for this difference is unclear. The assays in vitro use purified enzyme at a higher concentration than would be encountered in cells. Also, there may be differences in stability of PTP1B-OX in vitro compared to in cells. Resolution of this issue will require further study, including analysis of analogs of chelerythrine.

5. For all panels (Western blots), there needs to be a more thorough quantitative analysis including determination of statistical significance through replicate experiments.

This has been completed. Statistical analyses are now highlighted in the Figure Legends.

6. It is not clear whether changes in Fig 7D and E are statistically significant.

We feel that part of the problem here was the impression created by including protopine as a control. We have revised the figures to compare saline versus chelerythrine directly; the statistical analysis is now also highlighted in the Figure Legend. The data with protopine have now been moved to a new Supplementary Figure 15.

7. More than 20 single nucleotide polymorphisms (SNPs) that are associated with increased risk of type 2 diabetes have been identified within the PTPN1 gene---are these SNPs gain-of-function?

Although these SNPs may prove to be of interest, and would be a suitable subject for future study, at this time, to the best of our knowledge, they have not been fully characterized.

Reviewer #2 (Remarks to the Author):

Insulin signaling is critical for glucose homeostasis and the development of diabetes and its complications. The protein tyrosine phosphatase PTP1B occupies a critical position in signal transduction as a negative regulator of insulin and leptin signaling. PTP1B is reversibly oxidized and inactivated to allow the insulin receptor to be activated. Since knockout mice lacking PTP1B are viable and lean with improved glucose tolerance. Developing inhibitors of PTP1B is therefore of great interest to develop drugs against the epidemic of type II diabetes. Developing inhibitors of the active site in PTP1B has met with problems and as an alternative approach Tonks and coworkers have previously generated an antibody (scFv45) that binds to the oxidized inactive PTP1B and prevents its reductive activation. In this study they have characterized the binding mode of this antibody and identified a small molecule inhibitor of the PTP1B with the desired properties to only bind the oxidized PTP1B and not to PCPTP, which is a related phosphatase which can be associated with severe side effects if inhibited. They have also made extensive animal experiments to characterize the metabolic outcome in treated animals.

Overall this study deals with a highly important and interesting subject. The manuscript is of highest caliber and contains a wealth of data. The presentation is excellent. The extensive investigation and its results are highly significant and the paper presents a novel paradigm for inhibition of the enzyme PTP1B.

Reviewer #3 (Remarks to the Author):

Krishnan et al. mapped the site of interaction between oxidized PTP1B and scFv45 to a loop comprising residues 36-41 by mutagenesis and independently by structural proteomics. Substitution of three residues in this loop abolished the interaction without significant effects on catalytic activity. In silico docking indicated that acidic residues of ScFv45 have an important role in the interaction and that Lys41 in the interaction loop forms a hydrogen bond with Asp93 of ScFv45. A screen for compounds that lock oxidized PTP1B in its inactive conformation identified two candidate and a derivative, Chelerythrine, had even more pronounced effects. Docking of these compounds suggested a binding site overlapping the binding site of ScFv45 and Chelerythrine inhibited PTP1B-OX 15-fold more than K36AK41A PTP1B-OX.

Insulin signaling was enhanced by chelerythrine, which was abolished by catalase treatment. Leptin signaling induced PTP1B oxidation and chelerythrine treatment enhanced downstream Leptin signaling in tissue culture cells. Chelerythrine treatment improved metabolism in mice on a high fat diet. The authors conclude that their results provide proof-of-concept that stabilization of PTP1B in an inactive, oxidized conformation by small molecules can promote insulin and leptin signaling.

This is an interesting paper. In general, PTPs have proven difficult to target by inhibitors and this paper indicates a new mechanism to specifically inhibit PTP1B by locking the oxidized form in its inactive state.

Points:

1. The authors convincingly show that chelerythrine locks oxidized PTP1B in the inactive conformation. Is this interaction irreversible? Does prolonged incubation of chelerythrine-treated oxidized PTP1B with reducing agents lead to reduction and release of chelerythrine? Does chelerythrine somehow block reduction of the catalytic cysteine, e.g. by blocking access of reducing agents?

The effects of chelerythrine on PTP1B-OX are reversible. In a new Supplementary Figure 8, we show that phosphatase activity was restored by dilution of the PTP1B-OX-chelerythrine complex by 100-fold.

We have addressed this question through examining the effects of thioredoxin, which functions to reduce oxidized cysteines in target proteins. Thioredoxin contains a -CXXC- motif, in which the first cysteine forms a mixed disulphide intermediate with an oxidized cysteine in the target protein. The second cysteine then serves a “resolving” function – it forms a disulphide bond with the first cysteine in the -CXXC- motif, with the resulting transfer of electrons leading to reduction of the target protein. As summarized in new Supplementary Figures 11 and 12, we developed an assay using a “trapping mutant” form of thioredoxin in which the second cysteine is mutated to serine, which stabilizes the mixed disulfide intermediate with the target substrate.

In Supplementary Figure 11, we now show that the thioredoxin trapping mutant formed a complex with PTP1B only after treatment with H₂O₂, and that formation of this complex was not blocked by chelerythrine. This was seen reproducibly over three independent experiments. In Supplementary Figure 12, we show that this complex with the thioredoxin trapping mutant was formed by both wild-type and mut2 (chelerythrine-insensitive) forms of PTP1B. These data provide further evidence that the effects of chelerythrine are not simply due to blocking access of reducing agents.

2. Fig. 6. Chelerythrine has a modest effect on insulin signaling. Given that chelerythrine locks PTP1B in an inactive state, the most profound response will presumably become evident when the transient response to insulin declines at later time-points. Are the effects of chelerythrine more profound at later time points? Note that in the control, the 20 min timepoint shows the greatest response also in the control. Along the same lines, the response to leptin in the presence of ScFv45 (panel D) shows a plateau at 20, 30 and 60 min for the control and the ScFv45 expressing condition, which is surprising, because one would expect higher responses at later time points.

We agree that the effects of chelerythrine on insulin signaling are modest – in fact this is what you would expect from a compound that is close to a screening hit and that has yet to be fully optimized. We anticipate that subsequent SAR-based optimization would generate drug candidates with enhanced potency and efficacy. Also, we would anticipate that a stabilizer of PTP1B-OX would yield enhanced and sustained insulin signaling – we would expect to see effects both early and late in the signaling response. We would expect to see enhanced signaling at early time points because insulin-induced oxidation would attenuate the immediate inhibitory effect of PTP1B on the signaling response. Although we would

not necessarily expect to see higher maximal signaling, we would expect to see sustained signaling because insulin-induced oxidation would prevent the normal reactivation of PTP1B that is associated with signal termination.

To address this, we conducted an additional experiment to examine the effects of chelerythrine over an extended time course of treatment with insulin. As shown in a new Supplementary Figure 13, we observed that insulin, at a lower concentration than used originally (10nM), induced a transient autophosphorylation of the beta subunit of its receptor; however, in the presence of chelerythrine, this was both enhanced at early time points and sustained.

The difference between control and scFv45 expressing cells appears modest in panel D, yet quantification shows a 7-fold increase. This quantification appears to overestimate the difference. In order to present the quantitation in the most comprehensive manner, we have combined the data from three independent experiments rather than presenting quantitation of a single blot.

3. Discussion. Chelerythrine has been reported to act on other targets in cells, particularly PKC. Are the earlier described effects of chelerythrine on cells consistent with PTP1B inhibition?

We addressed the issue of other targets of chelerythrine in the Discussion. It has been reported that PKC may inhibit insulin signaling via phosphorylation of a Ser residue in IRS1 [Li et al, JBC (2004) 279:45304-45307]. Therefore, if chelerythrine were to inhibit PKC, this could also contribute to activation of insulin signaling. However, it is important to emphasize, as we discuss in the manuscript, that our data (including the observation that the effects of chelerythrine on insulin signaling were abrogated by expression of catalase) suggest that PKC is NOT the target of chelerythrine in this context. Furthermore, as also cited in our Discussion, the latest thinking within the PKC field, as expressed forcefully at a recent conference by one of its leading lights, Dr Alexandra Newton (UCSD), is that PKC is not a target of chelerythrine in vivo.

Minor points:

1. Fig. 3A. The numbers of the indicated residues in the Figure and the legend are not consistent.
2. Fig. 5D. Reduced and oxidized PTP1B are represented in different colors, gray and blue. Please indicate in the legend which is which.
3. Fig. 6D-G. For clarity, the authors should mention "leptin" as the factor that was added to the cells.
3. Fig. 7. The legends of panel A and B are switched. Legend to panel A indicates that oxidation of PTP1B was measured. How was this measured?
4. Supplementary Fig. 9. Indicate highlighted residue names and numbers in panel B.

These minor points have all been addressed as requested.

Reviewers' comments:

Reviewer #1 (Remarks to the Author):

The authors were responsive to many of my concerns. However they do need to provide data showing target engagement for the small molecule chelerythrine. I am not asking a proteome-wide unbiased assay using mass spectrometry. Minimally, they should show that the compound could engage PTP1B and not PKC in a cellular context, using specific antibodies to these proteins. Again, this is a crucial experiment which does not require any fancy equipment or expensive tools. The authors have all the reagents in place to conduct this experiment.

Reviewer #3 commented to the editor that you have satisfactorily addressed all his/hers comments and thus he/she was in support of the publication of your paper.

Response to Reviewers' comments:

Our response to the points raised by the referee is inserted in blue font.

Reviewer #1 (Remarks to the Author):

The authors were responsive to many of my concerns. However they do need to provide data showing target engagement for the small molecule chelerythrine. I am not asking a proteome-wide unbiased assay using mass spectrometry. Minimally, they should show that the compound could engage PTP1B and not PKC in a cellular context, using specific antibodies to these proteins. Again, this is a crucial experiment which does not require any fancy equipment or expensive tools. The authors have all the reagents in place to conduct this experiment.

We find this response rather frustrating. The referee requests that we “show that the compound could engage PTP1B and not PKC in a cellular context...”. Although in the early 1990s chelerythrine was reported to be an inhibitor of PKC, as you can see from the papers attached to this response, there is now a substantial body of data, from multiple labs, to illustrate that this is not the case. The situation is best summarized in the attached review from Alexandra Newton’s lab, which lists chelerythrine under the section “Discredited PKC inhibitors” on page 204-205 and states “... multiple independent investigators have debunked chelerythrine as a PKC inhibitor, both in vitro and in cells”. In addition, I have attached a couple of primary papers that make this point. Consequently, one frustration is that this referee is insisting that we address a non-existent problem!

Our own data already illustrate that the effects of chelerythrine are antagonized by expression of catalase, which serves to degrade hydrogen peroxide (Figure 6) - this supports our mechanism based upon stabilization of PTP1B-OX and is not consistent with a mechanism that is based upon inhibition of PKC. Now, to try and provide additional data to address the referee’s concern, we have added another experiment, included as Supplementary Figure 14. In this we show that when you ectopically expressed wild type PTP1B it antagonized insulin-induced phosphorylation of the beta subunit of the insulin receptor, and that this was counteracted by treatment with chelerythrine. In contrast, when you expressed PTP1B-mut2, which is catalytically competent, oxidized in response to insulin, but insensitive to chelerythrine, the compound was no longer able to restore beta

subunit phosphorylation. Again, this provides further support for PTP1B-OX as the target through which chelerythrine exerts its effects on insulin signaling in these cells.

Although he/she makes it sound trivial, the experiment that the referee is requesting is far from trivial. Notwithstanding the data demonstrating that chelerythrine is not a PKC inhibitor, the referee is asking for a negative result that could be open to several interpretations. Also, it is expecting a lot to demand that an interaction between the oxidized form of PTP1B and chelerythrine is maintained throughout the immunoprecipitation and washing steps.

Consequently, we hope that the further additional data provided in this second revision will be sufficient to address any concerns.

Reviewer #3 commented to the editor that you have satisfactorily addressed all his/hers comments and thus he/she was in support of the publication of your paper.

We are delighted that the referee supports publication.